# A toxin-mediated policing system in *Bacillus* optimizes division of labor via penalizing cheater-like nonproducers

Rong Huang[1], Jiahui Shao[1], Zhihui Xu[1], Yuqi Chen[1], Yunpeng Liu[2], Dandan Wang[3], Haichao Feng[1], Weibing Xun[1], Qirong Shen[1], Nan Zhang[1]*, Ruifu Zhang[1,2]*

[1]Jiangsu Provincial Key Lab of Solid Organic Waste Utilization, Jiangsu Collaborative Innovation Center of Solid Organic Wastes, Nanjing Agricultural University, Nanjing, China; [2]State Key Laboratory of Efficient Utilization of Arid and Semi-arid Arable Land in Northern China, Institute of Agricultural Resources and Regional Planning, Chinese Academy of Agricultura Sciences, Beijing, China; [3]National Engineering Research Center for Efficient Utilization of Soil and Fertilizer Resources, College of Resources and Environment, Shandong Agricultural University, Tai'an, China

**Abstract** Division of labor, where subpopulations perform complementary tasks simultaneously within an assembly, characterizes major evolutionary transitions of cooperation in certain cases. Currently, the mechanism and significance of mediating the interaction between different cell types during the division of labor, remain largely unknown. Here, we investigated the molecular mechanism and ecological function of a policing system for optimizing the division of labor in *Bacillus velezensis* SQR9. During biofilm formation, cells differentiated into the extracellular matrix (ECM)-producers and cheater-like nonproducers. ECM-producers were also active in the biosynthesis of genomic island-governed toxic bacillunoic acids (BAs) and self-resistance; while the nonproducers were sensitive to this antibiotic and could be partially eliminated. Spo0A was identified to be the co-regulator for triggering both ECM production and BAs synthesis/immunity. Besides its well-known regulation of ECM secretion, Spo0A activates acetyl-CoA carboxylase to produce malonyl-CoA, which is essential for BAs biosynthesis, thereby stimulating BAs production and self-immunity. Finally, the policing system not only excluded ECM-nonproducing cheater-like individuals but also improved the production of other public goods such as protease and siderophore, consequently, enhancing the population stability and ecological fitness under stress conditions and in the rhizosphere. This study provides insights into our understanding of the maintenance and evolution of microbial cooperation.

*For correspondence: nanzhang@njau.edu.cn (NZ); rfzhang@njau.edu.cn (RZ)

## Editor's evaluation

This manuscript reports notable findings regarding the potential for self-policing and a division of labor among biofilm-inhabiting Bacillus cells. Overall, this work is robust in its use of various techniques and provides solid insights into the intersections of well-understood regulatory controls and the suppression of cheaters. Colleagues interested in microbial social interactions should find this study's narrative about the internal mediation of cell differentiation valuable.

## Introduction

Cooperative interactions are not restricted to complex, higher organisms, but are also prevalent among microbial communities in many contexts (*Kehe et al., 2021*; *Rakoff-Nahoum et al., 2016*;

*Wenseleers and Ratnieks, 2006*). Both natural selection and game theory predict that cooperative systems are vulnerable to non-cooperative cheaters that exploit the benefit, such as public goods including extracellular enzymes (*Chen et al., 2019*), siderophore (*Griffin et al., 2004*), or biofilm matrix (*Dragoš and Kovács, 2017*; *Vlamakis et al., 2013*) since these selfish individuals enjoy the common resources without paying their cost (*Hardin, 1968*; *Martin et al., 2020*; *West et al., 2006*). Intriguingly, cooperation principally survives cheating during evolutionary history (*Travisano and Velicer, 2004*), and a couple of mechanisms have been proposed to play significant roles in maintaining cooperation by preventing cheater invasion (*Özkaya et al., 2017*; *Travisano and Velicer, 2004*). These strategies mainly include kin selection/discrimination (*Özkaya et al., 2017*; *Diggle et al., 2007*; *McNally et al., 2017*), facultative cooperation regulated by a quorum-sensing (QS) system (*Allen et al., 2016*), or nutrient fitness cost (*Sexton and Schuster, 2017*), coupling production of public and private goods (*Dandekar et al., 2012*), punishment of cheating individuals by cooperator-produced antibiotics (*García-Contreras et al., 2015*; *Wang et al., 2015*), partial privatization of public goods under certain conditions (*Jin et al., 2018*; *Otto et al., 2020*), and spatial structuring (*van Gestel et al., 2014*). In general, the emergence of multiple sanction mechanisms is a consequence of natural selection, which suppresses social cheaters and enhances the altruistic behavior, thereby maintaining microbial community stability and improving their adaptation in different niches (*Özkaya et al., 2017*).

In certain cases, microbial cooperation involves the division of labor, where subpopulations of cells are specialized to perform different tasks (*Dragoš et al., 2018a*; *Strassmann and Queller, 2011*). Division of labor requires three basic conditions: individuals exhibit different tasks (phenotypic variation); some individuals carry out cooperative tasks that benefit other individuals (cooperation); all individuals gain an inclusive fitness benefit from the interaction (adaptation) (*Dragoš et al., 2018b*; *West and Cooper, 2016*). For instance, *Bacillus subtilis* colony will phenotypically differentiate into surfactin-producing and matrix-producing cells during sliding motility, where the surfactin reduces the friction between cells and their substrate, while the matrix assembles into van Gogh bundles that drive the migration (*Jordi et al., 2015*). Another typical case is in early-stage biofilms, an extracellular matrix (ECM)-the enclosed multicellular community that sustains bacterial survival in diverse natural environments; it is known that *B. subtilis* cells can differentiate into motile cells and matrix-producing cells during biofilm formation (*Vlamakis et al., 2008*; *Chai et al., 2008*; *Shank and Kolter, 2011*; *López and Kolter, 2010*; *López et al., 2009c*; *López et al., 2009a*; *van Gestel et al., 2015*; *Vlamakis et al., 2013*; *Kearns, 2008*). The advantage of the division of labor is to efficiently integrate distinct cellular activities, thereby endowing a community with higher fitness than undifferentiated clones (*West and Cooper, 2016*; *Zhang et al., 2016*).

Importantly, efficient division of labor relies on elaborate coordination of cell differentiation (*Cremer et al., 2019*; *López et al., 2009b*; *Lord et al., 2019*). In relative to the subpopulation producing a certain public good (e.g. cells producing ECM during biofilm formation), the nonproducing cells that can also enjoy this common good, actually become the 'cheater-like' individuals to some extent (although they may provide other contributions to the community) (*Claessen et al., 2014*; *Otto et al., 2020*; *West and Cooper, 2016*). Therefore, regulating the proportion of each cell type and alignment of interests, is important for maintaining the stability and fitness of the division of labor (*West and Cooper, 2016*), while an unbalanced cell differentiation will reduce the population productivity and even cause a collapse of the division of labor (*Dragoš et al., 2018a*). Despite the knowledge of pathways controlling cell differentiation in microbes, little is known about how the different cell types interact with each other and the fitness consequences of their interaction (*van Gestel et al., 2015*). Although a few studies have investigated the overlap between public goods production and cell cannibalism (*González-Pastor et al., 2003*; *López et al., 2009c*), as well as matrix privatization (*Otto et al., 2020*) during cell differentiation, the molecular mechanism involved in coordinating the cheater-like individuals in the division of labor, as well as the ecological significance of the policing system in regulating population stability and fitness, remain unclear. Accordingly, lacking these knowledge limits our understanding of cooperation and altruism within microbial social communities.

*Bacillus velezensis* SQR9 (formerly *B. amyloliquefaciens* SQR9) is a well-studied beneficial rhizobacterium that forms robust and highly structured biofilms on the air-liquid interface and plant roots (*Qiu et al., 2014*; *Xu et al., 2019a*; *Xu et al., 2013*; *Cao et al., 2011*). Strain SQR9 harbors a novel genomic island 3 (GI3) consisting of four operons, where the second, third, and fourth operons are responsible for the production of the novel branched-chain fatty acids, BAs, while the first operon

encodes an ABC transporter to export toxic BAs for self-immunity (*Wang et al., 2019*). Production of toxic BAs was proved to occur in the subfraction of cells with the self-immunity ability induced by BAs during biofilm formation, where the nonproducing siblings will be lysed by BAs (*Huang et al., 2021*; *Wang et al., 2019*). Based on the manifestation that the BA-mediated cannibalism enhanced the biofilm formation of strain SQR9, we hypothesized the ECM and BAs synthesis can be co-regulated to restrain the cheater-like individuals that don't produce ECM, thereby optimizing the division of labor and altruistic behavior. Using a combination of single-cell tracking techniques, molecular approaches, and ecological evaluation, we demonstrated that ECM and BAs production are coordinated in the same subpopulation by the same regulator during biofilm formation, which enforces punishment of the cheater-like nonproducers to maintain community stabilization; also this genomic island-governed policing system is significant to promote community fitness in various conditions.

## Results

### Coordinated production of ECM and autotoxin BAs punishes cheater-like nonproducers in the *B. velezensis* SQR9 community

*Bacillus* cells in early-stage biofilms are known to contain specialized groups as motile cells and matrix-producing cells (*Kearns, 2008*; *van Gestel et al., 2015*; *Vlamakis et al., 2008*). We hypothesized that secretion of cannibal toxin BAs can eliminate ECM nonproducers in *B. velezensis* SQR9 biofilm, and try to determine the subpopulation for ECM (public goods) production and BAs (autotoxin) biosynthesis/BAs-induced self-immunity, as well as their interactions. We fused promoters for genes related to extracellular polysaccharides (EPS) and TasA fibers (two dominant ECM components in *Bacillus* biofilm *Vlamakis et al., 2013*) biosynthesis with *mCherry*, while the promoters for genes related to the autotoxin BAs biosynthesis and the self-immunity with *gfp*, obtained the transcriptional reporter $P_{eps}$-*mCherry*, $P_{tapA}$-*mCherry*, $P_{bnaF}$-*gfp*, and $P_{bnaAB}$-*gfp*, respectively. Their expression patterns were monitored using confocal laser scanning microscopy (CLSM) during the biofilm community formation. Photographs show that expression of the $P_{eps}$-*mCherry*, $P_{tapA}$-*mCherry*, $P_{bnaF}$-*gfp*, and $P_{bnaAB}$-*gfp* were all observed in a subpopulation cells of the whole community (*Figure 1*), which suggests a functional division of labor during biofilm formation; this cell differentiation pattern also indicates the ECM-nonproducers can be recognized as the cheater-like individuals (*Otto et al., 2020*). Importantly, the overlay of the double fluorescent reporters indicates that ECM and BAs production is generally raised in the same subpopulation (*Figure 1*; the yellow cells represent co-expression of *mCherry* and *gfp*), the flow cytometry also confirms the positive correlation between the two reporters within the picked cells as expected (*Figure 1—figure supplement 1*), since the self-immunity gene *bnaAB* was reported to be specifically activated by endogenous BAs (*Huang et al., 2021*), it was also preferentially expressed in the same subpopulation with ECM-producers (*Figure 1*, *Figure 1—figure supplement 1*). These results demonstrate general coordination of ECM production and BAs synthesis/immunity in the same subpopulation of the *B. velezensis* SQR9 biofilm community.

Based on the co-expression pattern, we postulated that the ECM-nonproducing cheater-like cells, synchronously being sensitive to the BAs, could be killed by their siblings that produce both public goods ECM and the autotoxin BAs. Combining propidium iodide (a red-fluorescent dye for labeling dead cells) staining with reporter labeling, we monitored the cell death dynamics during the biofilm formation process in real-time. It was observed that a portion of the cells that didn't produce public ECM or toxic BAs, or silenced in expression of the self-immunity gene *bnaAB* (cells without GFP signal), were killed by adjacent corresponding producers during the biofilm development process (*Figure 2*), while these producers remained alive throughout the incubation (*Figure 2—videos 1–4*); importantly, the number of dead cells adjacent to the producers was significantly higher than that closed to the non-producers (*Figure 2—figure supplement 1*). This lysis can be attributed to the BAs produced by the *gfp*-activated cells, as cannibalism of *B. velezensis* SQR9 was largely dependent on the production of this secondary metabolism (*Huang et al., 2021*). Taken together, the double-labeling observation and cell death dynamics detection indicate that the subpopulation of ECM and BAs producers selectively punish the nonproducing siblings, depending on a coordinately activated cell-differentiation pathway.

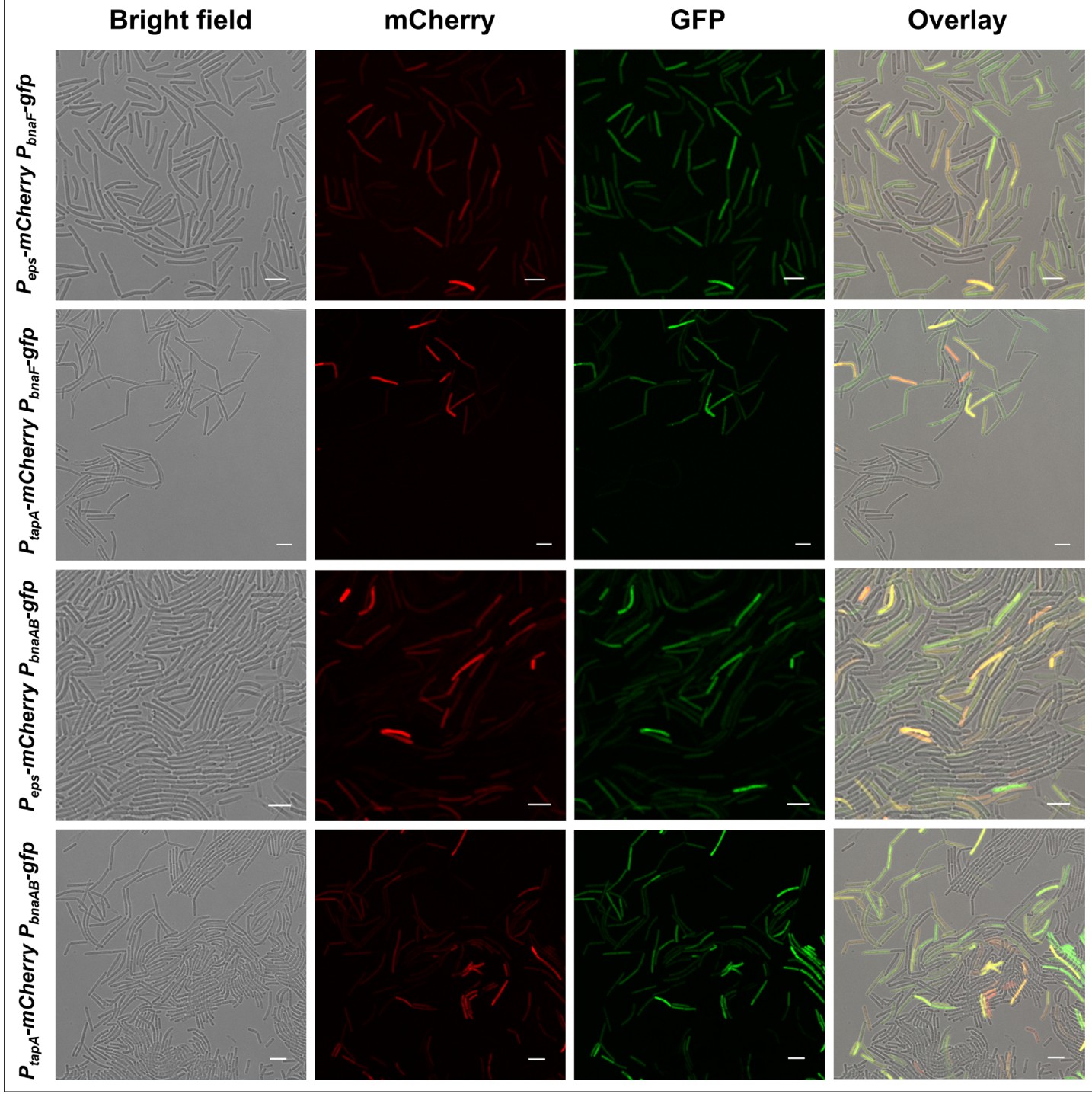

**Figure 1.** Expression of ECM production and BAs biosynthesis/immunity were located in the same subpopulation. Fluorescence emission patterns of double-labeled strains. Colony cells of different double-labeled strains were visualized using CLSM to monitor the distribution of fluorescence signals from different reporters. $P_{eps}$-*mCherry* and $P_{tapA}$-*mCherry* were used to indicate cells expressing extracellular polysaccharides (EPS) and TasA fibers production, respectively; $P_{bnaF}$-*gfp* and $P_{bnaAB}$-*gfp* were used to indicate cells expressing BAs synthesis and self-immunity, respectively. The bar represents 5 μm.

The online version of this article includes the following figure supplement(s) for figure 1:

**Figure supplement 1.** Quantification of fluorescence emission patterns of double-labeled strains.

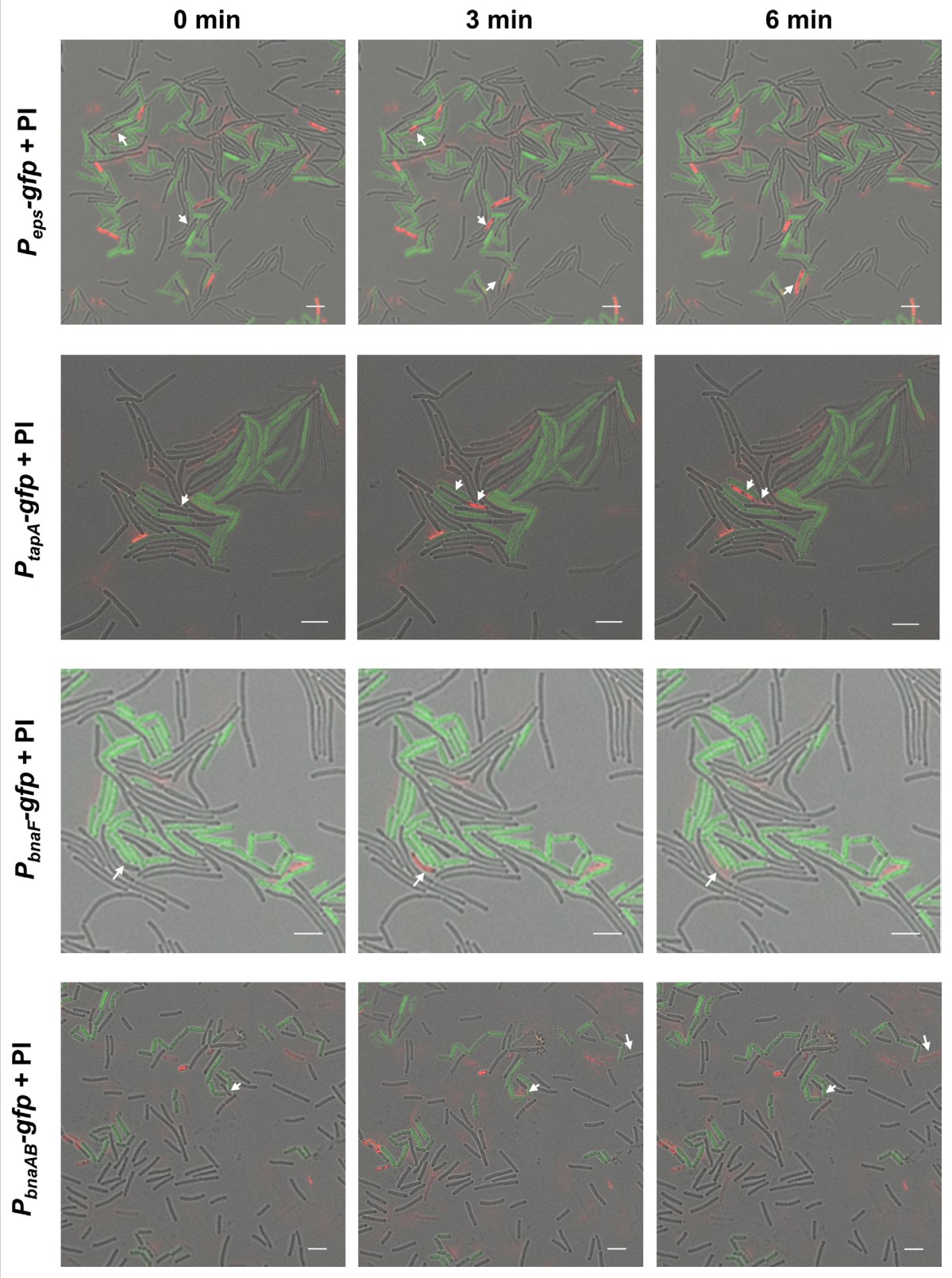

**Figure 2.** ECM and BAs producing subpopulations eliminated the nonproducing cheaters. The time-lapse experiment for observing the source and distribution of dead cells. Colony cells of different *gfp*-labeled strains were stained with propidium iodide (PI, a red-fluorescent dye for labeling dead cells) for 15 min, and then visualized by a CLSM to monitor the distribution of fluorescence signal from reporters and the PI dye. '0 min' represents the time point at which cells are alive as shown by the arrow, '3 min' or '6 min' is the time point afterward, and the cells at the arrow die or even break apart.

*Figure 2 continued on next page*

*Figure 2 continued*

$P_{eps}$-*gfp* and $P_{tapA}$-*gfp* were used to indicate cells expressing extracellular polysaccharides (EPS) and TasA fibers production, respectively; $P_{bnaF}$-*gfp* and $P_{bnaAB}$-*gfp* were used to indicate cells expressing BAs synthesis and self-immunity, respectively. The total number of cells is 198 for strain SQR9-$P_{eps}$-*gfp*, 71 for strain SQR9-$P_{tasA}$-*gfp*, 88 for strain SQR9-$P_{bnaF}$-*gfp*, and 162 for strain SQR9-$P_{bnaAB}$-*gfp*. The bar represents 5 µm.

The online version of this article includes the following video and figure supplement(s) for figure 2:

**Figure supplement 1.** Quantification of the source and location of dead cells newly appeared during 3 hr observation.

**Figure 2—video 1.** Dynamic observation of extracellular polysaccharides (EPS)-producing cells and dead cells in *B. velezensis* SQR9 community during biofilm formation.

https://elifesciences.org/articles/84743/figures#fig2video1

**Figure 2—video 2.** Dynamic observation of TasA fibers-producing cells and dead cells in *B. velezensis* SQR9 community during biofilm formation.

https://elifesciences.org/articles/84743/figures#fig2video2

**Figure 2—video 3.** Dynamic observation of BAs-producing cells and dead cells in *B. velezensis* SQR9 community during biofilm formation.

https://elifesciences.org/articles/84743/figures#fig2video3

**Figure 2—video 4.** Dynamic observation of BAs-immunity cells and dead cells in *B. velezensis* SQR9 community during biofilm formation.

https://elifesciences.org/articles/84743/figures#fig2video4

## Spo0A is the co-regulator for triggering ECM production and BAs synthesis/immunity

To identify the potential co-regulator(s) of ECM production and BAs synthesis/immunity in *B. velezensis* SQR9, we evaluated the BAs production in an array of mutants that are known to be altered in ECM synthesis (Δ*degU*, Δ*comPA*, Δ*abrB*, Δ*sinI*, Δ*sinR*, and Δ*spo0A*), by measuring their antagonism towards *B. velezensis* FZB42, a target strain specifically inhibited by BAs but no other antibiotics secreted by SQR9 (*Wang et al., 2019*). The BAs extract of wild-type SQR9 showed remarkable antagonism to the lawn of strain FZB42 (*Figure 3A and B*); only Δ*spo0A* but no other mutants (all with the equal cell density of the wild-type), revealed a significantly reduced inhibition zone towards FZB42, and the complementary strain generally restored the antagonistic ability (*Figure 3A and B*). Spo0A is a well-investigated master regulator that governs multiple physiological behaviors in *B. subtilis* and closely-related species *Hamon and Lazazzera, 2001*; *Molle et al., 2003*; *Xu et al., 2019b*; as expected, the EPS production and biofilm formation was seriously impaired in Δ*spo0A* (*Figure 3—figure supplement 1*). Intriguingly, Δ*spo0A* but neither its complementary strain nor the wild-type, can be substantially inhibited by the BAs extract of strain SQR9, while Δ*spo0A* was not inhibited by ΔGI3 that disabled in BAs production (*Figure 3C*), suggesting Spo0A does participate in the immunity to BAs. In addition, we constructed *gfp* transcriptional fusions to the promoter of genes involved in ECM production (*eps* & *tapA*) and BAs biosynthesis/immunity (*bnaF/bnaAB*) and discovered that under both liquid culture (*Figure 3D*) and plate colony conditions (*Figure 3—figure supplement 2*), their expression level was significantly decreased in Δ*spo0A* as compared with the wild-type, which was restored in the complementary strain Δ*spo0A/spo0A*. These results suggest that the global regulator Spo0A is the co-regulator for controlling ECM production and BAs biosynthesis/immunity in *B. velezensis*, which is probably dependent on the transcriptional regulation of certain relevant genes.

## Spo0A activates acetyl-CoA carboxylase (ACC) to support BAs synthesis and self-immunity

In *Bacillus*, Spo0A governs the regulatory pathway for matrix gene (the *eps* and *tapA-sipW-tasA* operons) expression by controlling the activity of the regulators SinR and AbrB (*Vlamakis et al., 2013*), but how it mediates BAs synthesis and self-immunity remains unknown. We used biolayer interferometry analysis (BLI) for detecting molecular interaction signals between protein and DNA fragments (an increased signal during association and a decreased signal during dissociation). Results showed that the purified protein Spo0A cannot directly bind to the promoter of *bnaF*, suggesting it doesn't induce BAs production through direct transcriptional activation (*Figure 4—figure supplement 1*). Alternatively, Spo0A has been reported to stimulate the expression of *accDA* that encodes ACC (*Diomandé et al., 2015*; *Pedrido et al., 2013*), which catalyzes acetyl-CoA to generate malonyl-CoA, an essential precursor for BAs biosynthesis (*Figure 4A*; *Wang et al., 2019*); therefore, we postulated *accDA* may be involved in the regulation of BAs production/immunity by Spo0A. We firstly

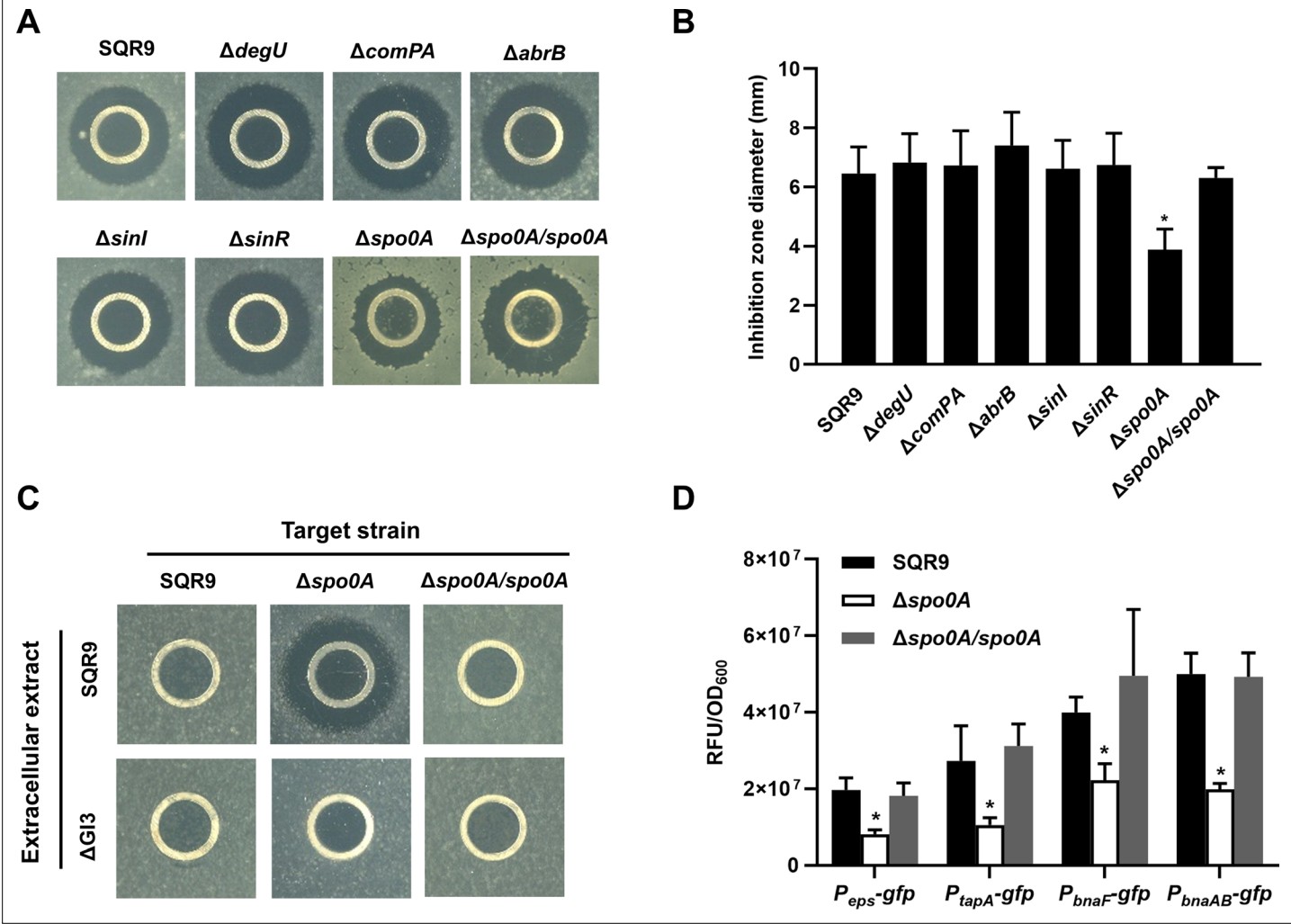

**Figure 3.** Spo0A is the co-regulator for triggering ECM production and BAs synthesis/immunity. (**A**) Oxford cup assay. Inhibition of the lawn of *B. velezensis* FZB42 by the BAs extract of wild-type SQR9, its different mutants altered in ECM production, and complementary strain Δ*spo0A/spo0A*. (**B**) Quantification of inhibition zone. Diameter of the inhibition zones is observed in (**A**). (**C**) Oxford cup assay. Sensitivity of wild-type SQR9, Δ*spo0A*, and Δ*spo0A/spo0A* (as the lawn) to the extracellular extract of SQR9 and its mutant ΔGI3 that disable BAs synthesis. (**D**) Quantification of fluorescence in liquid culture. The expression level of *eps*, *tapA*, *bnaF*, and *bnaAB* in wild-type SQR9, Δ*spo0A*, and Δ*spo0A/spo0A*, as monitored by using *gfp* reporters fused to the corresponding promoters. Data are means and standard deviations from three biological replicates. * indicates a significant difference with the Control (SQR9) column as analyzed by Student's *t*-test (p<0.05).

The online version of this article includes the following source data and figure supplement(s) for figure 3:

**Source data 1.** Related to *Figure 3B*.

**Source data 2.** Related to *Figure 3D*.

**Figure supplement 1.** The biofilm formation and EPS production were seriously impaired in Δ*spo0A*.

**Figure supplement 2.** Expression level of *eps*, *tapA*, *bnaF*, and *bnaAB* in the colony cells of wild-type SQR9, Δ*spo0A*, and Δ*spo0A/spo0A*, as monitored by using *gfp* reporters fused to the corresponding promoters.

verified the positive regulation of Spo0A on *accDA* expression in *B. velezensis* SQR9 by *gfp* fusion (*Figure 4B*, *Figure 4—figure supplement 2* ). Since knockout of *accDA*, the essential gene for fatty acids biosynthesis, significantly impacts bacterial growth, we alternatively constructed a strain in which the original promoter of *accDA* was replaced by a xylose-inducible promoter ($P_{xyl}$), and monitored its BAs synthesis/immunity under different xylose induction conditions. The SQR9-$P_{xyl}$-*accDA* lost the antagonism ability towards target strain FZB42 in the absence of xylose, while the inhibition was significantly enhanced with the induction of xylose in a dose-dependent manner (*Figure 4C and D*). Since exogenous xylose didn't influence the suppression of wild-type SQR9 on FZB42 (*Figure 4C*

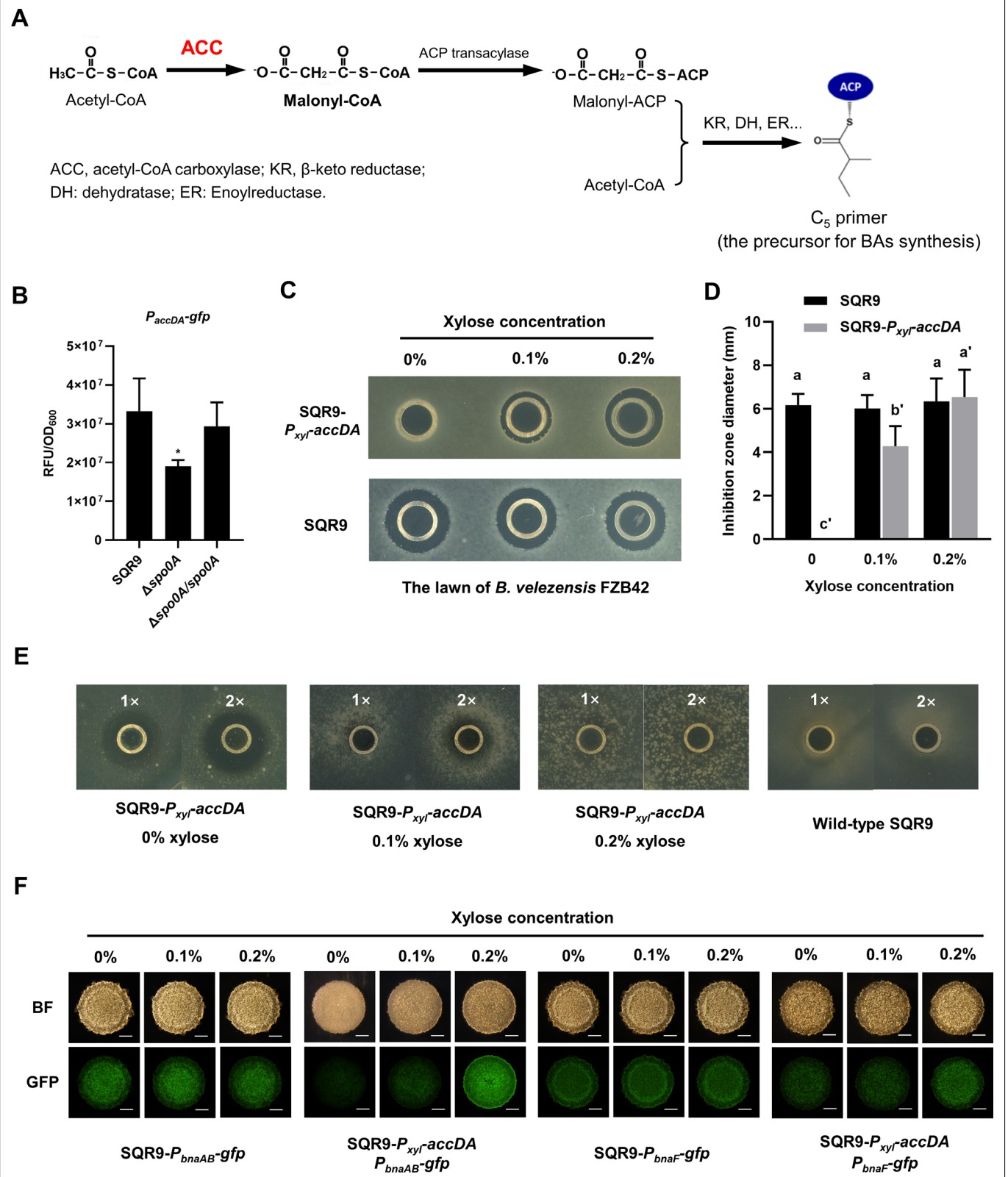

**Figure 4.** Spo0A activates ACC for BAs synthesis and self-immunity. (**A**) Involvement of ACC in the biosynthesis of BAs in *B. velezensis* SQR9. ACC catalyzes acetyl-CoA to generate malonyl-CoA, which is transformed to malonyl-ACP under the catalyzation of ACP transacylase; then malonyl-ACP and acetyl-CoA are aggregated into a $C_5$ primer, the precursor for BAs synthesis. (**B**) Quantification of fluorescence in liquid culture. The expression level of *accDA* in wild-type SQR9, Δ*spo0A*, and Δ*spo0A/spo0A*, as monitored by using the $P_{accDA}$-*gfp* reporter. (**C**) Oxford cup assay. Inhibition of the lawn of

*Figure 4 continued on next page*

*Figure 4 continued*

*B. velezensis* FZB42 by the BAs extract of wild-type SQR9 and SQR9-$P_{xyl}$-accDA, with the addition of different concentrations of xylose (0%, 0.1%, and 0.2%). (**D**) Quantification of inhibition zone. Diameter of the inhibition zones is observed in (**C**). (**E**) Oxford cup assay. Sensitivity of wild-type SQR9 and SQR9-$P_{xyl}$-accDA (as the lawn) to the BAs extract of SQR9 (100 μL (1x) or 200 μL (2x)), with the addition of different concentrations of xylose (0%, 0.1%, and 0.2%). (**F**) Colony fluorescence. Expression of *bnaF* and *bnaAB* in the colony cells of wild-type SQR9 and SQR9-$P_{xyl}$-accDA, with the addition of different concentrations of xylose (0%, 0.1%, and 0.2%). Colonies were observed under both bright fields (BF in the figure) and GFP channel, to monitor the fluorescence of $P_{bnaF}$–*gfp* and $P_{bnaAB}$-*gfp* reporters in different strains. The bar represents 1 mm. Data are means and standard deviations from three biological replicates. * in (**B**) indicates a significant difference (p<0.05) with the Control (SQR9) column as analyzed by Student's *t*-test; columns with different letters in (**D**) are statistically different according to Duncan's multiple range test ('a' for wild-type SQR9 under different concentrations of xylose and 'a''' for SQR9-$P_{xyl}$-accDA; p<0.05).

The online version of this article includes the following source data and figure supplement(s) for figure 4:

**Source data 1.** Related to *Figure 4B*.

**Source data 2.** Related to *Figure 4D*.

**Figure supplement 1.** Interaction between the purified protein Spo0A and the promoter of *bnaF* ($P_{bnaF}$) as determined by Biolayer interferometry data (BLI).

**Figure supplement 2.** Expression of *accDA* in the colony cells of wild-type SQR9, Δ*spo0A*, and Δ*spo0A/spo0A*, as monitored by using *gfp* reporters fused to the promoter of *accDA*.

**Figure supplement 3.** Quantification of the inhibition zone is shown in *Figure 4E*.

**Figure supplement 4.** Quantification of *bnaAB* and *bnaF* in strain SQR9 and SQR9-$P_{xyl}$-accDA.

**Figure supplement 5.** Production of ACC and BAs immunity were located in the same subpopulation.

*and D*), these results suggest that *accDA* expression positively contributes to BAs production. Importantly, the SQR9-$P_{xyl}$-accDA was proved to be sensitive to SQR9-produced BAs without xylose addition, and the immunity was gradually restored with xylose supplement (*Figure 4E*, *Figure 4—figure supplement 3*). The xylose-induced transcription of *accDA*, also resulted in enhanced expression of genes involved in self-immunity (*bnaAB*; *Figure 4F*, for quantitative intensity please see *Figure 4F*, *Figure 4—figure supplement 4A, C*), but not BAs synthesis (*bnaF*; *Figure 4F* and *Figure 4—figure supplement 4B, D*), as the AccDA-derived malonyl-CoA accumulation affects BAs production in a post-transcriptional manner. The CLSM photographs and flow cytometry analysis also reveal that the activation of *accDA* (*mCherry* fusion) and *bnaAB* (*gfp* fusion) was located in the same subpopulation cells (*Figure 4—figure supplement 5*). Accordingly, these results indicate the positive regulation of Spo0A on BAs production/immunity in *B. velezensis* SQR9, is strongly dependent on *accDA* that encodes ACC.

## The co-regulation policing system optimizes the division of labor and promotes population fitness

Having illustrated the molecular mechanism of the co-regulation pathway for punishing nonproducing cheater-like cells in *B. velezensis* SQR9, we wondered about the broad-spectrum ecological significance of this policing system for *B. velezensis* SQR9 at a community level. We constructed two mutants with disabled sanction mechanism, the Δ*bnaV* deficient in BAs synthesis (loss of the punishing weapon) and the SQR9-$P_{43}$-bnaAB that continually expresses the self-immunity genes (cheater-like individuals cannot be punished by the weapon BAs), both mutants showed similar growth characteristics with the wild-type (*Figure 5—figure supplement 1*). We first applied flow cytometry analysis to test whether the lack of the policing system (Δ*bnaV* and SQR9-$P_{43}$-bnaAB) impairs the punishment of public goods-nonproducers during biofilm formation. The proportion of matrix-producing cooperators (*eps* & *tapA* active cells) in the wild-type community, as well as the average expression level of corresponding genes, were significantly higher than that in the Δ*bnaV* or SQR9-$P_{43}$-bnaAB community (*Figure 5A and B*), suggesting the division of labor in the two mutants population was significantly different with the wild-type. Consequently, the wild-type established a more vigorous biofilm as compared with the two mutants, as shown by the earlier initial progress, larger maximum biomass, and delayed dispersal process (prolonged stationary phase) (*Figure 5C and D*). Additionally, the robust biofilm formed by the wild-type also endowed them with stronger resistance against different stresses, including antibiotics, salinity, acid-base, and oxidation (*Figure 5D*, *Figure 5—figure supplement 2* and *Figure 5—figure*

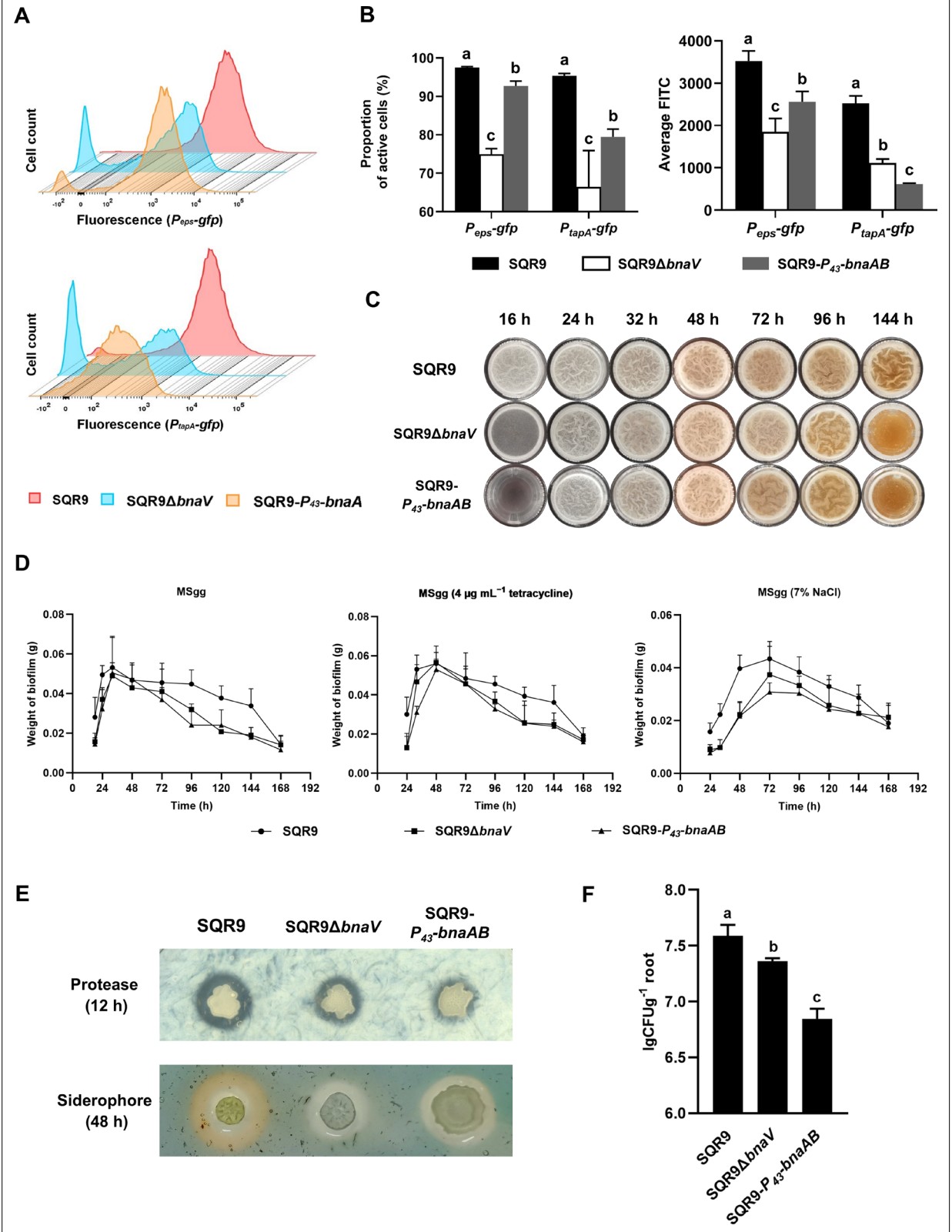

**Figure 5.** The co-regulation policing system optimizes the division of labor and enhances population fitness. (**A**) Flow cytometry monitoring the expression of $P_{eps}$-$gfp$ and $P_{tapA}$-$gfp$ reporters in wild-type SQR9, SQR9Δ$bnaV$, and SQR9-$P_{43}$-$bnaAB$. (**B**) Quantification of (**A**). The proportion of the active cells (%) and average FITC in wild-type SQR9, SQR9Δ$bnaV$, and SQR9-$P_{43}$-$bnaAB$, as monitored by $P_{eps}$-$gfp$ and $P_{tapA}$-$gfp$ reporters using flow cytometry. (**C**) Pellicle morphology. Pellicle formation dynamics of wild-type SQR9, SQR9Δ$bnaV$, and SQR9-$P_{43}$-$bnaAB$ in MSgg medium. (**D**) Quantification of

*Figure 5 continued on next page*

*Figure 5 continued*

pellicles. Pellicle weight dynamics of wild-type SQR9, SQR9Δ*bnaV,* and SQR9-*P₄₃-bnaAB* in MSgg medium under normal (corresponds to (**C**)) or stressed conditions (H$_2$O$_2$, tetracycline, or 7% NaCl). (**E**) Qualitative analysis of protease or siderophore yield. Production of proteases and siderophore by wild-type SQR9, SQR9Δ*bnaV,* and SQR9-*P₄₃-bnaAB* colonies. (**F**) Root colonization assay. Comparison of root colonization of wild-type SQR9, SQR9Δ*bnaV,* and SQR9-*P₄₃-bnaAB*. Data are means and standard deviations from three biological replicates; columns with different letters are significantly different according to Duncan's multiple range tests, p<0.05.

The online version of this article includes the following source data and figure supplement(s) for figure 5:

**Source data 1.** Related to *Figure 5B*.

**Source data 2.** Related to *Figure 5D*.

**Source data 3.** Related to *Figure 5F*.

**Figure supplement 1.** Growth curves of wild-type SQR9, SQR9Δ*bnaV,* and SQR9-*P₄₃-bnaAB*.

**Figure supplement 2.** Pellicle morphology.

**Figure supplement 3.** Quantification of pellicles.

**Figure supplement 4.** Quantification of neutral/alkaline protease activity and siderophore production by wild-type SQR9, SQR9Δ*bnaV,* and SQR9-*P₄₃*-*bnaAB*.

*supplement 3*). These data indicate the policing system in wild-type SQR9 ameliorates the division of labor during biofilm formation, thereby promoting community fitness.

Besides the well-known regulation of biofilm matrix production, Spo0A also controls the production of other public goods such as proteases and siderophore (*Fujita et al., 2005*; *Molle et al., 2003*); it can be recognized as a critical switch that governs the cell transition from a free-living and fast-growing status (Spo0A-OFF), to a multicellular and cooperative style (Spo0A-ON) (*Shank and Kolter, 2011*; *López et al., 2009c*). Intrinsically, the punishing targets of this policing system are supposed not limited to the cheater-like matrix-nonproducers, but all of the Spo0A-OFF individuals (cells that don't express the immune genes *bnaAB*, including protease-nonproducers and siderophore-nonproducers). Therefore, we determined the production of extracellular proteases and siderophore among the three strains, revealing that these public goods were also accumulated more in the wild-type than in these two mutants' communities (*Figure 5E*, *Figure 5—figure supplement 4*). Importantly, the wild-type SQR9 demonstrated significantly stronger root colonization compared with the two mutant strains losing the cheater punishing system (*Figure 5F*). In summary, the Spo0A governed co-regulation punishment system effectively optimizes the division of labor and altruistic behavior in *the B. velezensis* population, by excluding the cheater-like nonproducers to a certain degree, consequently improving the population stability and ecological fitness under different conditions.

## Discussion

Division of labor, where subpopulations perform complementary tasks simultaneously within an assembly, characterizes major evolutionary transitions of cooperation in certain cases (*Babak, 2018*). Unlike the diverse strategies for preventing obligate cheaters in cooperative systems (*Özkaya et al., 2017*; *Smith and Schuster, 2019*; *Travisano and Velicer, 2004*), division of labor requires an efficiency benefit and alignment of interests covering different specialized individuals (*West and Cooper, 2016*). For instance, compared with cells that produce a certain kind of public goods (e.g. ECM or extracellular hydrolases), the subpopulations that don't perform these tasks (but still share these benefits) become cheater-like individuals, and their proportion needs to be controlled for maintaining community stability and fitness (*Martin et al., 2020*; *West and Cooper, 2016*). In the present study, we demonstrated that during biofilm formation, the beneficial rhizobacterium *B. velezensis* SQR9 engages a policing system that coordinately actives ECM production and autotoxin synthesis/immunity, to punish the cheater-like subpopulation silencing in public goods secretion and restrain their proportion in the community (*Figure 6*). Importantly, the optimized division of labor not only facilitates ECM accumulation but also contributes to elevated production of other public goods including proteases and siderophore, thereby improving the community fitness under different stressful conditions and in plant rhizosphere (*Figure 5*), which could be defined as an effective strategy for enhancing cooperation and altruism. The coordination policing system suppresses subpopulation that stays in a fast-growing, motility phase (Spo0A-OFF state), to promote the population to a stationary, resource-mining phase

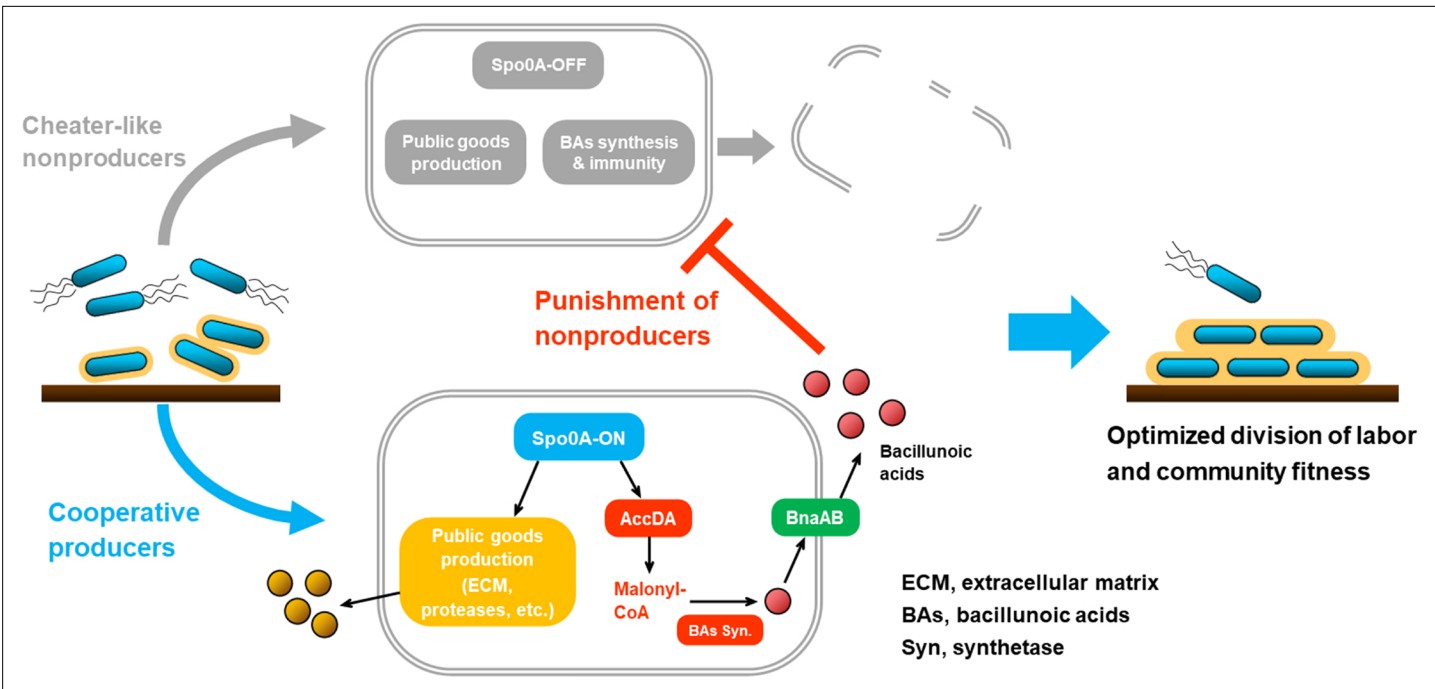

**Figure 6.** A working model and ecological significance of the co-regulation policing system in *B. velezensis*. In certain conditions (e.g. environmental or self-produced clues, surface attachments, etc.), *Bacillus* cells can differentiate into Spo0A-ON (~moderate phosphorylated) and Spo0A-OFF (unphosphorylated) subpopulations. The Spo0A-ON subpopulation is cooperators that produce public goods for the community, such as ECM or proteases; simultaneously they express AccDA to produce malonyl-CoA as the precursor for BAs biosynthesis, and the endogenous autotoxin activates immunity-required transporter BnaAB to pump them out. Comparatively, the Spo0A-OFF subpopulation is cheat-like individuals that are silenced in public goods secretion, which are also disabled in malonyl-CoA production and BAs biosynthesis/self-immunity. Consequently, the cooperators-produced BAs can effectively eliminate the cheater-like nonproducers, thereby optimizing the division of labor and enhancing population fitness.

(Spo0A-ON state) when the environment required (*Figure 6*). Our finding coincides with the phenomenon that Spo0A-dependent toxin killing of Spo0A-OFF cells in *B. subtilis* enhances biofilm formation and delays sporulation progress, which can be attributed to both eliminating of matrix-nonproducers and releasing of available nutrients (*González-Pastor et al., 2003*; *López et al., 2009c*; *Huang et al., 2021*). It should be noted that the coordination system for optimizing the division of labor is relatively temperate than those for excluding obligate cheaters (*Özkaya et al., 2018*; *Wang et al., 2015*), as only a subpopulation of the cheater-like individuals were killed (*Figure 2*); we think this scene is a balance between restraining the cheater-like subpopulation and retaining the advantages of cell differentiation (*Babak, 2018*; *Kaern et al., 2005*; *López et al., 2009c*).

The molecular working model of the present policing system, being Spo0A simultaneously regulates ECM production and also the toxin-antitoxin system (*Figure 6*), represents typical co-regulation machinery for mediating microbial social interactions (*Dandekar et al., 2012*; *van Gestel et al., 2015*; *Wang et al., 2015*). The opportunistic pathogen *Pseudomonas aeruginosa* engages the QS circuit (LasR-LasI and RhlI-RhlR systems) to couple the production of public and private goods for placing a metabolic restraint on cheaters (*Dandekar et al., 2012*; *Whiteley et al., 2017*); the *P. aeruginosa* cooperators can also punish LasR-null social cheaters by producing cyanide, where cooperators acquire immunity from the QS system while cheaters are sensitive to this toxin (*Wang et al., 2015*; *Yan et al., 2019*). During biofilm formation and sporulation by *B. subtilis*, the global regulator Spo0A simultaneously induces the production of matrix and cannibalism toxins (Skf and Sdp); since genes responsible for toxin synthesis and self-immunity are simultaneously expressed, the matrix producers can be resistant to these toxins while the sensitive nonproducers will be selectively penalized (*Ellermeier et al., 2006*; *González-Pastor et al., 2003*; *López et al., 2009c*). In the comparison of *B. subtilis* and *B. velezensis* SQR9, the similarity is that the global regulator Spo0A controls the synthesis of the ECM and the cannibalism toxin; the difference lies in the type of cannibalism toxin and its synthesis/regulation pathway. Specifically, the Spo0A-governed policing system in *B. velezensis*

SQR9 is extremely unique: (i) The toxic BAs for punishment are novel antimicrobial fatty acids that firstly identified in strain SQR9, which mediate cannibalism and strongly inhibit the growth of closely related *Bacillus* strains; also its synthesis is encoded by a horizontal gene transfer (HGT)-acquired genomic island (*Wang et al., 2019*). (ii) Spo0A doesn't mediate the BAs production/self-immunity in a direct transcriptional regulation way, but activates AccDA for accumulating the precursor for BAs biosynthesis (a post-transcriptional manner; *Figure 4*). The accumulated BA precursor may induce the expression of BA synthetase genes; additionally, the self-resistance is mainly induced by intra-cellular BAs through a two-component system (*Huang et al., 2021*). Therefore, the foreign genomic island and the indigenous Spo0A regulation pathways, constitute an ingenious coordination system for punishing cheater-like nonproducers and enhancing clonemate cooperation.

Relevant to how such a complex system could evolve, a possible scenario is a gradual evolution through transitional states. Perhaps homogeneous biofilm formation was the ancestral state, where all cells in the community are matrix producers (*Khare and Shaulsky, 2006*). Thereafter, heterogeneous biofilm raised as cells specialize in motile or matrix-producing subpopulations; it would be favored if the benefit of quickly responding to drop-in nutrients outweighed the cost of having cheater-like nonproducers that reduced the ability to form a biofilm (*Hamilton, 1964*; *Smits et al., 2006*; *López et al., 2009c*; *Joan et al., 2011*; *West and Cooper, 2016*). Furthermore, the heterogeneous biofilm strategy provides the evolutionary context of sanctioning behavior (*Acar et al., 2008*; *López et al., 2009c*; *West and Cooper, 2016*; *Spratt and Lane, 2022*). Interestingly, the genomic island responsible for BAs synthesis in *B. velezensis* SQR9 acquired through HGT, not only acts as a weapon for antagonizing closely related competitors (*Wang et al., 2019*), but also establishes a policing system for punishing cheater-like individuals within the biofilm community. Considering that bacterial biofilm is a major lifestyle in the natural environment (*Hall-Stoodley et al., 2004*), the dual ecological benefits probably explain why this large cluster was integrated into the genome of strain SQR9; also this case can provide inspiration for discovering novel molecular regulatory mechanisms and understanding microbial evolution events (*Strassmann et al., 2000*). Alternatively, this sanction system can work in concert with a privatization strategy to collectively enhance cooperation during biofilm formation (*Otto et al., 2020*).

In conclusion, the present study highlights the beneficial rhizobacterium *B. velezensis* SQR9 engages a policing system that coordinately actives ECM production and autotoxin synthesis/immunity, to penalize the cheater-like subpopulation silencing in public goods secretion, thereby enhancing the division of labor and community fitness. This study provides insights into the molecular mechanism involved in controlling cell differentiation, as well as the ecological roles of the policing system, which deepens our understanding of the maintenance and evolution of microbial cooperation and altruistic behavior.

# Materials and methods

**Key resources table**

| Reagent type (species) or resource | Designation | Source or reference | Identifiers | Additional information |
|---|---|---|---|---|
| Strain, strain background (*Bacillus velezensis*) | SQR9 | Lab strain | CGMCC accession No. 5808 | |
| Strain, strain background (*Bacillus velezensis*) | FZB42 | *Chen et al., 2007* | BGSC accession no. 10A6 | |
| Strain, strain background (*Escherichia coli*) | Top 10 | Invitrogen | | Host for plasmids |
| Strain, strain background (*Escherichia coli*) | BL21 (DE3) | Invitrogen | | For recombinant protein expression |
| Recombinant DNA reagent | pNW33n (plasmid) | *Zhou et al., 2018* | | *B. subtilis-E. coli* shuttle vector |
| Gene (*Bacillus velezensis*) | spo0A | GenBank | V529_25300 | |
| Gene (*Bacillus velezensis*) | bnaA | GenBank | V529_06410 | |
| Gene (*Bacillus velezensis*) | bnaB | GenBank | V529_06420 | |

*Continued on next page*

*Continued*

| Reagent type (species) or resource | Designation | Source or reference | Identifiers | Additional information |
|---|---|---|---|---|
| Gene (*Bacillus velezensis*) | *bnaV* | GenBank | V529_06620 | |
| Software, algorithm | FlowJo V10 | FlowJo V10 | | |
| Software, algorithm | SPSS | SPSS | | |
| Other | Propidium iodide | Invitrogen | L7012 | (20 mM) |

## Bacterial strains and growth conditions

The strains and plasmids used in this study are listed in *Supplementary file 1a*. *Bacillus velezensis* SQR9 (formerly *B. amyloliquefaciens* SQR9, China General Microbiology Culture Collection Center (CGMCC) accession no. 5808) was used throughout this study. *B. velezensis* FZB42 (*Bacillus* Genetic Stock Center (BGSC) accession no. 10A6) was used to test the BAs production by wild-type SQR9 and its mutants. *Escherichia coli* TOP 10 (Invitrogen, Shanghai, China) was used as the host for all plasmids. *E. coli* BL21 (DE3) (Invitrogen, Shanghai, China) was used as the host for recombinant protein expression. All strains were routinely grown at 37 °C in low-salt Luria-Bertani (LLB) medium (10 g $L^{-1}$ peptone, 5 g $L^{-1}$ yeast extract, 3 g $L^{-1}$ NaCl). For biofilm formation, *B. velezensis* SQR9 and its mutants were cultivated in MSgg medium (5 mM potassium phosphate, 100 mM morpholine propanesulfonic acid, 2 mM $MgCl_2$, 700 µM $CaCl_2$, 50 µM $MnCl_2$, 50 µM $FeCl_3$, 1 µM $ZnCl_2$, 2 mM thiamine, 0.5% glycerol, 0.5% glutamate, 50 µg of tryptophan per milliliter, 50 µg of phenylalanine per milliliter, and 50 µg of threonine per milliliter) at 37 °C (*Branda et al., 2001*). To collect the fermentation supernatant for antagonism assessment, *B. velezensis* SQR9 and its mutants were cultured in Landy medium (*Landy et al., 1947*) containing 20 g $L^{-1}$ glucose and 1 g $L^{-1}$ yeast extract. When necessary, antibiotics were added to the medium at the following final concentrations: zeocin, 20 µg $mL^{-1}$; spectinomycin, 100 µg $mL^{-1}$; kanamycin, 30 µg $mL^{-1}$; ampicillin, 100 µg $mL^{-1}$; chloramphenicol, 5 µg $mL^{-1}$ for *B. velezensis* strains and 12.5 µg $mL^{-1}$ for *E. coli* strains; erythromycin, 1 µg $mL^{-1}$ for *B. velezensis* strains and 200 µg $mL^{-1}$ for *E. coli* strains. The medium was solidified with 2% agar.

## Reporter construction

For single-labeled strain, the promoter region of the testing gene and *gfp* fragment were fused through overlap PCR, and this transcriptional fusion was cloned into vector pNW33n using primers listed in *Supplementary file 1b*. For double-labeled strains, one promoter region was fused with *gfp* fragment, and the other promoter region was fused with the *mCherry* fragment. The two fusions were then fused in opposite transcription directions and cloned into vector pNW33n using primers listed in *Supplementary file 1b*. All constructions were transferred into competent cells of *B. velezensis* SQR9 and mutants when required.

## Promoter replacement

Strain SQR9-$P_{xyl}$-*accDA* was constructed by replacing the original promoter of *accDA* ($P_{accDA}$) with a xylose-inducible promoter $P_{xyl}$. The approximately 800 bp fragments upstream and downstream of the $P_{accDA}$ region were amplified from the genomic DNA of strain SQR9; the $Spc^r$ fragment was amplified from plasmid P7S6 (*Feng et al., 2018*), and the $P_{xyl}$ promoter was amplified from the plasmid PWH1510 (*Xu et al., 2019a*). The four fragments were fused using overlap PCR in the order of the upstream fragment, $Spc^r$, $P_{xyl}$, and the downstream fragment. The fusion was transferred into competent cells of *B. velezensis* SQR9 for generating transformants. Strain SQR9-$P_{43}$-*bnaAB* was obtained by replacing the original promoter ($P_{bnaAB}$) with a constitutive promoter $P_{43}$. The primers used for constructing the four-fragment fusion are listed in *Supplementary file 1b*.

## Fluorescence microscopy

Cells were inoculated from a fresh pre-culture and grown to mid-exponential growth at 37 °C in an LLB medium. Bacterial cultures were centrifuged at 4000 × g for 5 min, the pellets were washed and suspended in liquid MSgg to reach an $OD_{600}$ of 1.0. One µL suspension was placed on a solid MSgg medium and was cultured at 37 °C for 12 h. Agarose MSgg pads were then inverted on a glass bottom dish (Nest). Cells were imaged using the Leica TCS SP8 microscope with the 63x oil-immersion

objective lens. For GFP observation, the excitation wavelength was 488 nm and the emission wavelength was 500~560 nm; for mCherry observation, the excitation wavelength was 587 nm and the emission wavelength was 590~630 nm. Wild-type biofilms containing no fluorescent fusions were analyzed to determine the background fluorescence. The number of cells emitting mCherry, GFP, or both fluorescence was also collected for calculating the proportion; each treatment includes six biological replicates.

For the time-lapse experiment, after staining with propidium iodide (PI) for 15 min, images of early-stage biofilms on the agarose pad were recorded for 3 hr, with an interval of 3 min. Image acquisitions were also performed with the Leica TCS SP8 microscope with the 63x oil-immersion objective lens. Detectors and filter sets for monitoring of GFP and PI (excitation wavelength of 536 nm and emission wavelength of 608~652 nm) were used.

## Flow cytometry

Biofilms of 16 hr were collected and re-suspended in 1 mL PBS buffer, and single cells were obtained after mild sonication. Cells were centrifuged at 4000 × g for 5 min and washed briefly with PBS. For flow cytometry, cells were diluted to 1:100 in PBS and measured on BD FACSCanto II. For GFP fluorescence, the laser excitation was 488 nm and coupled with 500–560 nm.

For assessment of the double-labeled strains, cells were diluted to 1:100 in PBS and measured on BD FACS Symphony SORP. For GFP fluorescence, the laser excitation was 488 nm coupled with 530/30 and 505LP sequential filters; for mCherry fluorescence, the laser excitation was 561 nm coupled with 610/20 and 600LP sequential filters.

Every replicate was analyzed for 20,000 events. FlowJo V10 software was used for data analysis and graph creating. Three replicates were analyzed for each treatment.

## Preparation of the BAs extract

The BAs extract was prepared by thin layer chromatography (TLC). According to a previous study (*Wang et al., 2019*), the fermentation supernatant of strain SQR9 was separated on a TLC plate, and the inhibition zone on the lawn of strain FZB42 indicated the position of BAs. Then, silica gel powder with BAs was scraped and extracted by MeOH, which was used as the BAs extract.

## Oxford cup assay

Inhibition of different SQR9-derived mutants on *B. velezensis* FZB42 was evaluated by the Oxford cup method. The suspension of strain FZB42 (~$10^6$ CFU mL$^{-1}$) was spread onto LLB plates (10 × 10 cm) to grow as a bacterial lawn. A volume of 100 μL BAs extract produced by different mutants was injected into an Oxford cup on the lawn of strain FZB42. The plates were placed at 22 °C until a clear zone formed around the cup, and the inhibition diameter was scored. Each treatment includes three biological replicates.

## BAs-sensitivity assessment

Cells were inoculated from a fresh pre-culture and grown to mid-exponential growth at 37 °C in an LLB medium. Afterward, diluted cell suspension (~$10^6$ CFU mL$^{-1}$) was spread onto LLB plates to grow as a bacterial lawn. A volume of 100 μL BAs extracts from the wild-type SQR9 was injected into an Oxford cup on the lawn. The plates were placed at 22 °C for observation and determination of the inhibition zone. Each treatment includes three biological replicates.

## Biolayer interferometry (BLI) measurements

To confirm whether Spo0A can bind $P_{bnaF}$ directly, determination of binding kinetics was performed on an Octet RED96 device (ForteBio, Inc, Menlo Park, US) at 25 °C with orbital sensor agitation at 1000 rpm. Streptavidin (SA) sensor tips (ForteBio) were used to immobilize 100 nM biotin-labeled $P_{bnaF}$. Then, a baseline measurement was performed in the buffer PBST (PBS, 0.1% BSA, 0.02% Tween-20) for 300 s. The binding of Spo0A at different concentrations (100 nM, 250 nM, 500 nM, and 1000 nM) to $P_{bnaF}$ was recorded for 600 s followed by monitoring protein dissociation using PBST for another 600 s. The BLI data for each binding event was summarized as an 'nm shift' (the wavelength/spectral shift in nanometers) and KD values were determined by fitting to a 1:1 binding model.

## Promoter activity testing via fluorescence intensity

For colony fluorescence, cells were inoculated from a pre-culture into a fresh LLB medium and grown at 37 °C with 170 rpm shaking until $OD_{600}$ reached 0.5. One µL of the suspension was inoculated on a solid LLB medium and cultured at 37 °C. Colony morphology and fluorescence were recorded by the stereoscope. ImageJ software was used to measure GFP intensity. For liquid culture fluorescence, overnight cultures were transferred to a fresh LLB medium. Fluorescence intensity was determined by a microtiter plate reader. Each treatment includes three biological replicates.

## Xylose induction assay

For the xylose-induced BAs production assay, 30 µL overnight culture of SQR9-$P_{xyl}$-$accDA$ or wild-type SQR9 was transferred respectively into 3 mL fresh LLB liquid with different concentrations of xylose (0%, 0.1%, and 0.2%) and incubated at 37 °C, 170 rpm for 24 hr. Cell suspensions were adjusted to the same $OD_{600}$ and were centrifuged at 12,000 × g for 1 min. The cell-free supernatant was mixed with MeOH (volume ratio 2:1) to extract BAs. A volume of 100 µL extract was injected into an Oxford cup on the lawn of strain FZB42 (as described above). The plates were placed at 22 °C.

For the xylose-induced self-immunity assay, strain SQR9-$P_{xyl}$-$accDA$ was grown in LLB without xylose for 24 hr. The cell suspension was spread onto LLB plates containing different concentrations of xylose (0%, 0.1%, and 0.2%) to grow as the lawn. A volume of 100 µL (1x) or 200 µL (2x) BAs extract from the wild-type SQR9 was injected into an Oxford cup on the lawn. The plates were placed at 22 °C.

For xylose-induced gene expression assay, cells were inoculated from a pre-culture into fresh LLB medium with different concentrations of xylose (0%, 0.1%, and 0.2%), and were grown at 37 °C with 170 rpm shaking until $OD_{600}$ reached 0.5. One µL of suspension was inoculated on a solid LLB medium and was cultured at 37 °C, colony morphology and fluorescence were recorded by the stereoscope.

Each treatment in these assays includes three biological replicates.

## Biofilm formation

Cells were inoculated from a fresh pre-culture and grown to mid-exponential growth at 37 °C in an LLB medium. Bacterial cultures were centrifuged at 4000 × g for 5 min, the pellets were washed and suspended in MSgg medium to an $OD_{600}$ of 1.0. For colony observation, 1 µL of suspension was inoculated on a solid MSgg medium and cultured at 37 °C, then the colony morphology was recorded by the stereoscope. For pellicle observation, the suspension was inoculated into MSgg medium with a final concentration of 1% in a microtiter plate well, and the cultures were incubated at 37 °C without shaking.

Besides, the ability of the strain to form biofilm under stress was measured in the 48-well microtiter plate according to the method described above. When required, reagents that simulate stress were supplemented in the MSgg medium before inoculating, including oxidative stress (0.0025% $H_2O_2$), salt stress (7% NaCl), acid stress (pH 5), alkaline stress (pH 8), and antibiotic stress (4 µg mL$^{-1}$ tetracycline or 20 µg mL$^{-1}$ streptomycin). The amount of reagent added was determined according to a concentration gradient in the pre-experiment, and a concentration was chosen to inhibit wild-type growth without killing it. At different stages of biofilm development (initiation, progress, maturity, and dispersal), the MSgg medium underneath the biofilm was carefully removed by pipetting, and then the biofilm was taken and weighed.

Each treatment includes three biological replicates.

## Root colonization assay in hydroponic culture

The bacterial suspension was inoculated into 1/4 Murashige-Skoog medium to make the final $OD_{600}$ value to be 0.1, into which sterile cucumber seedlings with three true leaves were immersed. After being cultured with slowly shaking for two days, cells colonized on cucumber roots were determined by plate colony counting. In detail, roots were washed eight times in PBS to remove free and weakly attached bacterial cells. After vortexing for 5 min, until colonized bacteria were detached from roots, 100 µL of the bacterial suspension was plated onto LLB agar plates for quantification. Each treatment includes three biological replicates.

## Measurement of public goods production

Qualitative measurement of proteases production was done by inoculating 1 μL of bacterial suspension on a solid 2% skim milk medium and cultured at 30 °C until a transparent zone formed around colonies; quantitative measurements of alkaline protease and neutral protease activity were conducted according to a previous study (*Kunitz, 1947*). Qualitative and quantitative measurements of siderophore production were based on the universal chemical assay described by *Schwyn and Neilands, 1987*. Each treatment includes three biological replicates.

## Acknowledgements

This work was financially supported by the National Natural Science Foundation of China (31870096, 42090064, 31972512, 32072665, and 32072675), the National Key R&D Program of China (2022YFD1901300), the Fundamental Research Funds for the Central Universities (KYZZ2022003), and the National Key Research and Development Program (2021YFD1900300).

## Additional information

### Funding

| Funder | Grant reference number | Author |
| --- | --- | --- |
| National Natural Science Foundation of China | 31870096 | Ruifu Zhang |
| National Natural Science Foundation of China | 42090064 | Qirong Shen |
| National Natural Science Foundation of China | 31972512 | Zhihui Xu |
| National Natural Science Foundation of China | 32072665 | Nan Zhang |
| National Natural Science Foundation of China | 32072675 | Weibing Xun |
| National Key Research and Development Program of China | 2022YFD1901300 | Nan Zhang |
| Fundamental Research Funds for the Central Universities | KYZZ2022003 | Ruifu Zhang |
| National Key Research and Development Program of China | 2021YFD1900300 | Weibing Xun |

The funders had no role in study design, data collection and interpretation, or the decision to submit the work for publication.

### Author contributions

Rong Huang, Software, Validation, Visualization, Methodology, Writing – original draft; Jiahui Shao, Resources, Investigation; Zhihui Xu, Writing - review and editing; Yuqi Chen, Investigation; Yunpeng Liu, Haichao Feng, Weibing Xun, Validation; Dandan Wang, Methodology; Qirong Shen, Conceptualization; Nan Zhang, Conceptualization, Funding acquisition, Validation, Writing – original draft, Writing - review and editing; Ruifu Zhang, Conceptualization, Funding acquisition, Writing - review and editing

### Author ORCIDs

Rong Huang ⬦ http://orcid.org/0000-0003-1347-841X
Zhihui Xu ⬦ http://orcid.org/0000-0002-3987-8836
Nan Zhang ⬦ http://orcid.org/0000-0001-8444-7456

**Decision letter and Author response**
Decision letter https://doi.org/10.7554/eLife.84743.sa1
Author response https://doi.org/10.7554/eLife.84743.sa2

## Additional files

### Supplementary files
• Supplementary file 1. Strains, plasmids, and primers used in this study. (a) List of strains and plasmids used in this study. (b) List of primers used in this study.
• MDAR checklist

### Data availability
All data generated or analysed during this study are included in the manuscript and supporting file.

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
