## [Editor Report]

This manuscript reports notable findings regarding the potential for self-policing and a division of labor among biofilm-inhabiting Bacillus cells. Overall, this work is robust in its use of various techniques and provides solid insights into the intersections of well-understood regulatory controls and the suppression of cheaters. Colleagues interested in microbial social interactions should find this study's narrative about the internal mediation of cell differentiation valuable.

---

## [Decision Letter]

**Decision letter after peer review:**

Thank you for submitting your article "A toxin-mediated policing system in Bacillus optimizes division of labor via penalizing cheater-like nonproducers" for consideration by *eLife*. Your article has been reviewed by 3 peer reviewers, one of whom is a member of our Board of Reviewing Editors, and the evaluation has been overseen by Aleksandra Walczak as the Senior Editor. The following individual involved in review of your submission has agreed to reveal their identity: Bin Ni (Reviewer #2).

Essential revisions:

Addressing the following list of major suggestions is essential for substantiating the manuscript's conclusions. (Note: The reviewing editor has lightly edited the reviewers' statements for clarity.)

1) The microscopy needs additional quantitative analysis.

Item #(1) All of the microscopy needs total cell count numbers.

Item #(2) Although Figure 1 depicts the co-expression pattern of ECM production (eps and tapA) and policing system (bnaF and bnaA) as an individual-level property, it lacks statistical analysis and a spatial correlation test. The authors should provide flow-cytometry data that quantifies the overlap between different subpopulations; with this data, one could understand the cell differentiation pattern from a population perspective.

Item #(3) For the data presented in Figure 1, the authors need a more concrete discussion of methods and could expand upon their existing quantification to improve the understanding of the co-expression of EPS and BA genes. How were cells segmented? What were the thresholds for whether a cell emitted fluorescence, and how was such a threshold chosen? The authors should perform a more sophisticated colocalization/correlation analysis. e.g., if the cells have a high Peps signal, does that correlate with a high PbnaF signal?

Item #(4) In the CLSM photographs (figure supplement 6 and lines 195 – 199), only one frame and no population sizes were provided. Additional frames and population counts are needed.

2) The gfp/mCherry fusions need transcriptional validation. For example, use qPCR to ascertain the relationship between the transcription of the reporter genes and the genes (eps, tapA, etc.) to which they correspond.

3) Figure 2 is not definitive.

#(1) The PI staining microscopy (as shown in Figure 2 and the supplemental videos) does not demonstrate that only non-expressing cells perish. Also, the presented data shows many non-labeled cells without PI; why do some nearby non-gfp-expressing cells remain alive? Consider using a third reporter that is a marker for all living cells (constitutive expression).

Question #(2) How do the authors determine whether the PI-stained cells were producers or cheaters before they died based on the data shown in Figure 2? If a Peps-GFP cell dies, I presume it loses its GFP signal? Do the authors identify the cell types before death? Furthermore, in the figure legend, the authors should define what they mean by "near" an EPS- producing cell. Is this quantitatively defined?

Item #(3) Figure 2 shows no visual evidence that the distance to producer cells matters. Also, the counts for "Statistics" in Figure 2 lack total population sizes, error bars, and statistical analysis; these data are needed to determine whether "selectively punish" is an accurate conclusion or whether other cellular factors could also explain the PI-stained cells.

3) The authors need to standardize their measurements of gene expression. Currently, the authors determine gene expression in different ways. For example, in Figure 3D, the ratio between fluorescence intensity and cell density (RFU/OD600) is used to show the transcription of target genes. While in Figure 4F, only RFU is shown but OD600 is ignored. Using only the fluorescence data may not reflect the objective gene expression pattern if the xylose concentrations can affect the growth of wild-type SQR9 or the constructed strains.

4) The spo0A complementation is a partial rescue, most visible in cells with the eps promoter driving gfp expression. The authors need to address this discrepancy and discuss its implications.

5) The data in Figure 4E should be quantified, as similar data are in other figures.

*Reviewer #1 (Recommendations for the authors):*

– Figure 2 shows no visual evidence that the distance to producer cells matters. Also, the counts for "Statistics" in Figure 2 lack total population sizes, error bars, and statistical analysis; these data are needed to determine whether "selectively punish" is an accurate conclusion or whether other cellular factors could also explain the PI-stained cells.

– In general, the total cell count numbers (in all microscopy) are needed. I could not fully validate the imaging conclusions with the data and images provided.

– The spo0A complementation is a partial rescue, most visible in cells with the eps promoter driving gfp expression. The authors need to address this discrepancy and discuss its implications.

– I could not verify the conclusions from the CLSM photographs (figure supplement 6 and lines 195 – 199) as only one frame and no population sizes were provided.

– *B. subtilis* is a well-studied organism. It would be helpful for the authors to include how the pathways in this Bacillus strain compare to the known ones in *B. subtilis*. As written, it seems that accDA might be a direct regulator, but it's not clear from the results shown. (I took away that it's an upstream regulator, but is it indirect?)

– For all photography, explicitly state that the image represents ## experiments (in the legend).

– The supplemental videos do not show more than the still images and are not needed for this manuscript.

*Reviewer #2 (Recommendations for the authors):*

Although the manuscript is concise and brings a detailed study of social interactions within microbial populations, I nevertheless have a few major concerns. Some of my main concerns would require additional experiments and a re-write of some part of the manuscript.

1) The co-expression pattern of ECM production (eps and tapA) and policing system (bnaF and bnaA) was shown in Figure 1 in an individual-level property, but lacks statistical analysis or spatial correlation test. I suggest authors provide a flow-cytometry data that quantifies the overlap between different subpopulations, thus we can understand the cell differentiation pattern in a population perspective.

2) The gfp/mCherry fusions need a transcriptional validation, such as the correspondence between the transcription of the reporter genes and corresponding genes (eps, tapA, etc.) determined by qPCR.

3) Determination of gene expression are performed in different ways. For example, in Figure 3D, the ratio between fluorescence intensity and cell density (RFU/OD600) is used to show the transcription of target genes. While in Figure 4F, only RFU is shown but OD600 is ignored; if the xylose concentrations can affect the growth of wild-type SQR9 or the constructed strains, using only the fluorescence data may not reflect the objective gene expression pattern.

4) According to the working model raised by the authors, it seems that expression of ECM-related genes (eps and tapA) and BAs-synthesis genes (bnaF) are not directly relevant, since Spo0A enhances ECM production in a transcriptional-regulation pattern but activates BAs biosynthesis in a post-transcriptional manner (by promoting the accumulation of the precursor). Interestingly, the CLSM graph shows that expression of these two sorts of genes is also likely to be overlapped. Authors should discuss or comment on this phenomenon in the manuscript.

5) Line 111-114, one additional sentence is needed here for introducing why the autotoxin system is involved in this scene.

6) Line 261, the reference citation format is incorrect.

The two references in Table S1 (Chen et al., 2007; Zhou et al., 2017) are missing in the Reference List.

7) Figures need to be standardized, such as font size and initial capitalization in Figure 5D.

8) There are a couple of grammatical and semantic errors in the manuscript. Please carefully go through the manuscript to correct them.

*Reviewer #3 (Recommendations for the authors):*

1. For the data presented in Figure 1, the authors need more concrete discussion of methods and could expand upon their existing quantification to improve the understanding of the co-expression of EPS and BA genes. How were cells segmented? What were the thresholds for whether a cell emitted fluorescence or not, and how was such a threshold chosen? The authors should perform a more sophisticated colocalization/correlation analysis. e.g. if the cells have high Peps signal, does that correlate with high PbnaF signal?

2. For the data presented in Figure 2, how do the authors determine that the PI-stained cells were producers or cheaters before death? If a Peps-GFP cell dies, I presume it loses its GFP signal? Do the authors identify the cell types before death? Furthermore, in the figure legend, the authors should define what they mean by "near" an EPS- producing cell. Is this quantitatively defined?

3. The data in Figure 4E should be quantified, as similar data are in other figures.

---

## [Author Response]

Essential revisions:Addressing the following list of major suggestions is essential for substantiating the manuscript's conclusions. (Note: The reviewing editor has lightly edited the reviewers' statements for clarity.)

Thank you very much for sorting out and summarizing the problems existing in the manuscript, we have carefully addressed all these comments to improve this manuscript.

1) The microscopy needs additional quantitative analysis.Item #(1) All of the microscopy needs total cell count numbers.Item #(2) Although Figure 1 depicts the co-expression pattern of ECM production (eps and tapA) and policing system (bnaF and bnaA) as an individual-level property, it lacks statistical analysis and a spatial correlation test. The authors should provide flow-cytometry data that quantifies the overlap between different subpopulations; with this data, one could understand the cell differentiation pattern from a population perspective.Item #(3) For the data presented in Figure 1, the authors need a more concrete discussion of methods and could expand upon their existing quantification to improve the understanding of the co-expression of EPS and BA genes. How were cells segmented? What were the thresholds for whether a cell emitted fluorescence, and how was such a threshold chosen? The authors should perform a more sophisticated colocalization/correlation analysis. e.g., if the cells have a high Peps signal, does that correlate with a high Pbnf signal?Item #(4) In the CLSM photographs (figure supplement 6 and lines 195 – 199), only one frame and no population sizes were provided. Additional frames and population counts are needed.

Thank you very much for summarizing the problems of missing quantitative analysis from the microscopic observations. We have performed additional experiments and analyses to address all these comments. Please see the specific responses as follows.

Item #(1) The statistical analyses in the original Figure 1 were removed. A more accurate and objective method, flow cytometry, was used to analyze the fluorescence expression patterns of different double-labeled strains (revised Figure 1—figure supplement 1). The total cell number observed for each strain was 20,000, and this information is included in the caption of revised Figure 1—figure supplement 1.

The statistical analyses in the original Figure 2 were also removed. We re-observed cells in biofilms for 3 hours (revised Figure 2-video 1-4), the source and distribution of newly emerged dead cells during this period was counted and analyzed in detail (revised Figure 2—figure supplement 1). During the 3 hours, pictures covering a 6 minutes stage that show the elimination process of several non-*gfp* expressing cells, are selected for display in revised Figure 2; the total number of cells in this figure is 198 for strain SQR9-*P_eps_*-*gfp*, 71 for strain SQR9-*P_tasA_*-*gfp*, 88 for strain SQR9-*P_bnaF_*-*gfp*, and 162 for strain SQR9-*P_bnaAB_*-*gfp*, and this information is included in the caption.

Item #(2) We further monitored the expression pattern of mCherry and GFP signals in each single double-labeled cell by flow cytometry. The results revealed that both fluorescence was generally located in the same subpopulation in the community (revised Figure 1—figure supplement 1), which confirmed the positive correlation between the two reporters within the picked cells from a population perspective. Thus, ECM production (*eps* and *tapA*) and the policing system (*bnaF* and *bnaA*) are co-expressed.

Item #(3) The statistical analysis in original Figure 1 was replaced by flow cytometry data that can accurately access the fluorescence emitted by each cell of double-labeled strains and determine their distribution pattern (revised Figure 1—figure supplement 1). Flow cytometry showed that in all double-labeled strains, mCherry fluorescence intensity generally correlated with GFP fluorescence intensity. For example, in strain *P_eps_*-*mCherry P_bnaF_*-*gfp*, the *P_eps_* signal was correlated with the *P_bnaF_* signal to a certain extent, highlighting the co-expression of EPS and BA synthesis genes in the same subpopulation (revised Figure 1—figure supplement 1). With regards to the experimental method of flow cytometry, (1) mild sonication of the biofilms was performed to segment cells for flow cytometry analysis; (2) the wild-type strain expressing no fluorescence protein was used as a negative control; (3) the negative control establishes the background fluorescence of the experimental samples and is used to set the threshold by adjusting baseline PMT (photomultiplier tube) voltages of the instrument.

For CLSM observation, the wild-type strain without fluorescent proteins was used as a negative control to estimate the fluorescence threshold. Besides, early-stage biofilms with monolayer cells were selected for observation.

Item #(4) Additional frames have been provided (revised Figure 4—figure supplement 5A) and the fluorescence expression patterns of 20,000 cells has been analyzed by flow cytometry (revised Figure 4—figure supplement 5B). In detail, the fluorescent signals of both mCherry and GFP were detected to be in the same subpopulation of the double-labeled strain *P_accDA_-mCherry P_bnaAB_-gfp* (revised Figure 4—figure supplement 5B). Thus, the activation of *accDA* and *bnaAB* was located in the same subpopulation cells. Accordingly, the “Statistics” of the original figure was replaced by the flow cytometry analysis.

2) The gfp/mCherry fusions need transcriptional validation. For example, use qPCR to ascertain the relationship between the transcription of the reporter genes and the genes (eps, tapA, etc.) to which they correspond.

Thank you very much for listing this comment. We performed a qPCR validation and showed that the reporter gene expression was correlated with the corresponding genes (*epsD*, *tasA*, *accD*, *bnaF*, and *bnaA*).

**Author response image 1. sa2fig1:** 

3) Figure 2 is not definitive.#(1) The PI staining microscopy (as shown in Figure 2 and the supplemental videos) does not demonstrate that only non-expressing cells perish. Also, the presented data shows many non-labeled cells without PI; why do some nearby non-gfp-expressing cells remain alive? Consider using a third reporter that is a marker for all living cells (constitutive expression).Question #(2) How do the authors determine whether the PI-stained cells were producers or cheaters before they died based on the data shown in Figure 2? If a Peps-GFP cell dies, I presume it loses its GFP signal? Do the authors identify the cell types before death? Furthermore, in the figure legend, the authors should define what they mean by "near" an EPS- producing cell. Is this quantitatively defined?Item #(3) Figure 2 shows no visual evidence that the distance to producer cells matters. Also, the counts for "Statistics" in Figure 2 lack total population sizes, error bars, and statistical analysis; these data are needed to determine whether "selectively punish" is an accurate conclusion or whether other cellular factors could also explain the PI-stained cells.

Thank you very much for summarizing the problems existed in Figure 2. We have addressed all these comments carefully. Please see the specific responses as follows.

Item #(1) According to the reviewer's suggestion, an observation covering more complete biofilm forming process, as well as a more convinced data statistics, should be performed. We then re-conducted microscope observation lasting for 3 h during biofilm formation, and assess the source and location of dead cells for statistical analysis. The results showed that all dead cells were originated from the subpopulation that didn't express the *gfp* (the nonproducers), and the number of dead cells adjacent to the producers was significantly higher than that closed to the non-producers (please see the pictures in Figure 2 and revised Figure 2—figure supplement 1).

In addition, regarding the survival of some non-*gfp*-expressing cells near the producers, based on several relevant literatures^1-3^ and the observation in the present study, we speculate that the coordination system for optimizing the division of labor is relatively temperate, thus only a part of the nonproducers (relative sensitive cells or facing higher concentrations of the toxin) are eliminated. We think this scene is a balance between restraining the cheater-like subpopulation and retaining the advantages of cell differentiation.

Item #(2) The SQR9-*P_eps_*-*gfp* cells emitting GFP signal are supposed to be producers because their EPS synthesis gene is activated; conversely, cells that don’t emit GFP signal are nonproducing cheaters. Based on this criterion, we determined the source of dead cells in the newly captured time-lapse images. The results showed that all dead cells were from non-*gfp* expressing cells (revised Figure 2—figure supplement 1), while the active *P_eps_*-*gfp* cells kept alive and didn’t lose their GFP signal during the cannibalism progress. Additionally, the distribution of dead cells in the newly captured time-lapse images were also analyzed. We define "near" an EPS- producing cell as cells that were in contact with an EPS-producing cell, also this information has been included in the figure legend. The number of dead cells adjacent to producers was significantly higher than that of non-adjacent cells (revised Figure 2—figure supplement 1).

Item #(3) We re-observed the cell differentiation and live-dead distribution in early-stage biofilms for 3 hours (revised Figure 2-video 1-4). The progress of individual nonproducers from alive to initial death and even disappearance in a biofilm population is shown in Figure 2 of the revised manuscript; also the source (from *gfp*-expressing or non-expressing cells) and distribution (near GFP-emitting cells or silent cells) of the dead cells in the newly captured time-lapse images were statistically analyzed. The results showed that all dead cells were from non-*gfp* expressing cells, and the number of dead cells adjacent to producers (in contact with a *gfp* expressing cell) was significantly higher than that of non-adjacent cells (revised Figure 2—figure supplement 1). We think the re-collected data can demonstrate the "selectively punish" in the biofilm population of strain SQR9. Also, total population sizes, error bars, and statistical analysis have been provided in revised Figure 2—figure supplement 1.

3) The authors need to standardize their measurements of gene expression. Currently, the authors determine gene expression in different ways. For example, in Figure 3D, the ratio between fluorescence intensity and cell density (RFU/OD600) is used to show the transcription of target genes. While in Figure 4F, only RFU is shown but OD600 is ignored. Using only the fluorescence data may not reflect the objective gene expression pattern if the xylose concentrations can affect the growth of wild-type SQR9 or the constructed strains.

Thank you very much for this suggestion and sorry for missing the cell density. The ratio between fluorescence intensity and cell density (RFU/OD_600_) has been provided to show the transcription of target genes in the xylose-induced strain (SQR9-*P_xyl_*-*accDA*) during liquid culture. The xylose-induced transcription of *accDA* resulted in enhanced expression of genes involved in self-immunity (revised Figure 4—figure supplement 4C) and BAs synthesis (revised Figure 4—figure supplement 4D).

4) The spo0A complementation is a partial rescue, most visible in cells with the eps promoter driving gfp expression. The authors need to address this discrepancy and discuss its implications.

Thank you for the comments. We have sequenced gene *spo0A* and its promoter of the complementation strain, ensuring the sequence is correct and it indeed inserted into the *amyE* site of strain SQR9 chromosome. Thus, for unknown reasons, ex-situ replacement of gene *spo0A* might result a slight impairment as compared with the wild-type.

In addition, the fluorescent intensity emitted by the strain Δ*spo0A/spo0A*-*P_eps_*-*gfp* has been rechecked for several times, and the expression level was significantly enhanced compared with the original data and close to the wild-type (revised Figure 3—figure supplement 2).

5) The data in Figure 4E should be quantified, as similar data are in other figures.

Sorry for lack of quantification of data in Figure 4E. We have supplemented the statistical information (Duncan's multiple rang tests) in the revised Figure 4—figure supplement 3.

Reviewer #1 (Recommendations for the authors):– Figure 2 shows no visual evidence that the distance to producer cells matters. Also, the counts for "Statistics" in Figure 2 lack total population sizes, error bars, and statistical analysis; these data are needed to determine whether "selectively punish" is an accurate conclusion or whether other cellular factors could also explain the PI-stained cells.

Thank you for your suggestions. We re-observed the cell differentiation and live-dead distribution in early-stage biofilms for 3 hours (revised Figure 2-video 1-4). The progress of individual nonproducers from alive to initial death and even disappearance in a biofilm population is shown in Figure 2 of the revised manuscript; also the source (from *gfp*-expressing or non-expressing cells) and distribution (near GFP-emitting cells or silent cells) of the dead cells in the newly captured time-lapse images were statistically analyzed. The results showed that all dead cells were from non-*gfp* expressing cells, and the number of dead cells adjacent to producers (in contact with a *gfp* expressing cell) was significantly higher than that of non-adjacent cells (revised Figure 2—figure supplement 1). We think the re-collected data can demonstrate the "selectively punish" in the biofilm population of strain SQR9. Also, total population sizes, error bars, and statistical analysis have been provided in revised Figure 2—figure supplement 1.

– In general, the total cell count numbers (in all microscopy) are needed. I could not fully validate the imaging conclusions with the data and images provided.

Thank you for your comments. The statistical analyses in the original Figure 1 were removed. A more accurate and objective method, flow cytometry, was used to analyze the fluorescence expression patterns of different double-labeled strains (revised Figure 1—figure supplement 1). The total cell number observed for each strain was 20,000, and this information is included in the caption of revised Figure 1—figure supplement 1.

The statistical analyses in the original Figure 2 were also removed. We re-observed cells in biofilms for 3 hours (revised Figure 2-video 1-4), the source and distribution of newly emerged dead cells during this period was counted and analyzed in detail (revised Figure 2—figure supplement 1). During the 3 hours, pictures covering a 6 minutes stage that show the elimination process of several non-*gfp* expressing cells, are selected for display in revised Figure 2; the total number of cells in this figure is 198 for strain SQR9-*P_eps_*-*gfp*, 71 for strain SQR9-*P_tasA_*-*gfp*, 88 for strain SQR9-*P_bnaF_*-*gfp*, and 162 for strain SQR9-*P_bnaAB_*-*gfp*, and this information is included in the caption.

– The spo0A complementation is a partial rescue, most visible in cells with the eps promoter driving gfp expression. The authors need to address this discrepancy and discuss its implications.

Thank you for your comments. We have sequenced gene *spo0A* and its promoter of the complementation strain, ensuring the sequence is correct and it indeed inserted into the *amyE* site of strain SQR9 chromosome. Thus, for unknown reasons, ex-situ replacement of gene *spo0A* might result a slight impairment as compared with the wild-type.

In addition, the fluorescent intensity emitted by the strain Δ*spo0A/spo0A*-*P_eps_*-*gfp* has been rechecked for several times, and the expression level was significantly enhanced compared with the original data and close to the wild-type (revised Figure 3—figure supplement 2).

– I could not verify the conclusions from the CLSM photographs (figure supplement 6 and lines 195 – 199) as only one frame and no population sizes were provided.

Thank you for your comments. Additional frames have been provided (revised Figure 4—figure supplement 5A) and the fluorescence expression patterns of 20,000 cells has been analyzed by flow cytometry (revised Figure 4—figure supplement 5B). In detail, the fluorescent signals of both mCherry and GFP were detected to be in the same subpopulation of the double-labeled strain *P_accDA_-mCherry P_bnaAB_-gfp* (revised Figure 4—figure supplement 5B). Thus, the activation of *accDA* and *bnaAB* was located in the same subpopulation cells. Accordingly, the “Statistics” of the original figure was replaced by the flow cytometry analysis.

– *B. subtilis* is a well-studied organism. It would be helpful for the authors to include how the pathways in this Bacillus strain compare to the known ones in B. subtilis. As written, it seems that accDA might be a direct regulator, but it's not clear from the results shown. (I took away that it's an upstream regulator, but is it indirect?)

Thank you for your suggestions. In *B. subtilis*, as a global regulator, Spo0A simultaneously induces the production of extracellular matrix and toxic peptides (Skf and Sdp, sporulation killing factor and sporulation delaying protein, respectively)^7^. Thus, in the comparison of *B. subtilis* and *B. velezensis* SQR9, the similarity is that the global regulator Spo0A controls the synthesis of extracellular matrix and the cannibalism toxin. The difference lies in the type of cannibalism toxin and its synthesis/regulation pathway (Line 278~281 of the revised Discussion). For *B. subtilis*, Spo0A directly activate the peptide Skf synthesis genes *skfABH*. The first gene, *skfA*, encodes the 56-amino acid long pro-antimicrobial peptide, which is post-translationally modified by SkfB, a radical S-adenosyl-methionine enzyme^8,9^. The resulting pre-SkfA is further processed to its active state by the putative thioredoxin oxidoreductase SkfH^10^. In contrast to the directly controlling of *skf* expression, Spo0A only plays an indirect role for regulating the *sdp* locus by repressing the global regulator AbrB that inhibits *sdp* expression^11^. Upon transcription of the *sdpABC* operon, the toxic peptide SdpC is post-translationally modified by SdpAB to its active 63-amino acids-long form.

For *B. velezensis* SQR9, the global regulator Spo0A activates acetyl-coA carboxylase AccDA through a direct manner, and this regulation is also conserved in *B. subtilis*. AccDA catalyzes the production of malonyl-CoA, which is essential for the synthesis of BAs precursors. As a result, AccDA is not a direct regulator for BAs synthesis, but works as a critical enzyme that controls the synthesis of BAs precursors.

– For all photography, explicitly state that the image represents ## experiments (in the legend).

Thank you for your suggestion and sorry for unclearness. In the revised legends, each image is annotated with the experiment the image represents.

– The supplemental videos do not show more than the still images and are not needed for this manuscript.

Sorry for unclearness. The original videos provide little new information as compared with the corresponding figures. New supplemental videos shown in the revised version lasted longer period (3 hours) (revised Figure 2-video 1-4); specifically, pictures covering a 6 minutes stage that show the elimination process of several non-*gfp* expressing cells, are selected for display in revised Figure 2.

Reviewer #2 (Recommendations for the authors):Although the manuscript is concise and brings a detailed study of social interactions within microbial populations, I nevertheless have a few major concerns. Some of my main concerns would require additional experiments and a re-write of some part of the manuscript.

Thank you very much for your positive comments. The detailed response to your recommendations and suggestions are as follows.

1) The co-expression pattern of ECM production (eps and tapA) and policing system (bnaF and bnaA) was shown in Figure 1 in an individual-level property, but lacks statistical analysis or spatial correlation test. I suggest authors provide a flow-cytometry data that quantifies the overlap between different subpopulations, thus we can understand the cell differentiation pattern in a population perspective.

Thank you very much for your constructive suggestion. We further monitored the expression pattern of mCherry and GFP signals in each single double-labeled cell by flow cytometry. The results revealed that both fluorescence was generally located in the same subpopulation in the community (revised Figure 1—figure supplement 1), which confirmed the positive correlation between the two reporters within the picked cells from a population perspective. Thus, ECM production (*eps* and *tapA*) and the policing system (*bnaF* and *bnaA*) are co-expressed.

2) The gfp/mCherry fusions need a transcriptional validation, such as the correspondence between the transcription of the reporter genes and corresponding genes (eps, tapA, etc.) determined by qPCR.

Thank you for your suggestion. We performed a qPCR validation and showed that the reporter gene expression was correlated with the corresponding genes (*epsD*, *tasA*, *accD*, *bnaF*, and *bnaA*).

3) Determination of gene expression are performed in different ways. For example, in Figure 3D, the ratio between fluorescence intensity and cell density (RFU/OD600) is used to show the transcription of target genes. While in Figure 4F, only RFU is shown but OD600 is ignored; if the xylose concentrations can affect the growth of wild-type SQR9 or the constructed strains, using only the fluorescence data may not reflect the objective gene expression pattern.

Thank you very much for your comments and sorry for missing the cell density. The ratio between fluorescence intensity and cell density (RFU/OD_600_) has been provided to show the transcription of target genes in the xylose-induced strain (SQR9-*P_xyl_*-*accDA*) during liquid culture. The xylose-induced transcription of *accDA* resulted in enhanced expression of genes involved in self-immunity (revised Figure 4—figure supplement 4C) and BAs synthesis (revised Figure 4—figure supplement 4D).

4) According to the working model raised by the authors, it seems that expression of ECM-related genes (eps and tapA) and BAs-synthesis genes (bnaF) are not directly relevant, since Spo0A enhances ECM production in a transcriptional-regulation pattern but activates BAs biosynthesis in a post-transcriptional manner (by promoting the accumulation of the precursor). Interestingly, the CLSM graph shows that expression of these two sorts of genes is also likely to be overlapped. Authors should discuss or comment on this phenomenon in the manuscript.

Thank you very much for your comments. According to the manner that Spo0A controls ECM and BA synthesis, that expression of ECM-related genes (*eps* and *tapA*) and BAs-synthesis genes (*bnaF*) may not be directly relevant. Interestingly, both CLSM and flow cytometry data show that the expression of these two sorts of genes is also likely to be overlapped. It's common for a substrate or precursor to induce functional enzyme expression^12^, thus we speculated that the accumulated BA precursor (catalyzed by AccDA under the activation of Spo0A) may induce expression of BA synthetase genes, thus exhibiting such expression pattern. We have added this discussion in the revised manuscript (Line 287~288).

5) Line 111-114, one additional sentence is needed here for introducing why the autotoxin system is involved in this scene.

Thank you for your suggestion. One sentence as "We hypothesized that secretion of cannibal toxin BAs can eliminate ECM nonproducers in *B. velezensis* SQR9 biofilm" has been added in the revised manuscript (Line 109~111).

6) Line 261, the reference citation format is incorrect.The two references in Table S1 (Chen et al., 2007; Zhou et al., 2017) are missing in the Reference List.

Sorry for the mistakes. We have corrected the reference citation format in the revised manuscript and added references in the revised Supplementary File 1.

7) Figures need to be standardized, such as font size and initial capitalization in Figure 5D.

Thanks for your comments and sorry for the mistakes. We have carefully gone through all figures to standardize font size and initial capitalization, and have made correction in Figure 5D and Figure 5—figure supplement 3 of the revised version.

8) There are a couple of grammatical and semantic errors in the manuscript. Please carefully go through the manuscript to correct them.

Thank you for your comments and sorry for these errors. We have carefully gone through the whole manuscript to revise grammatical and semantic errors.

Reviewer #3 (Recommendations for the authors):1. For the data presented in Figure 1, the authors need more concrete discussion of methods and could expand upon their existing quantification to improve the understanding of the co-expression of EPS and BA genes. How were cells segmented? What were the thresholds for whether a cell emitted fluorescence or not, and how was such a threshold chosen? The authors should perform a more sophisticated colocalization/correlation analysis. e.g. if the cells have high Peps signal, does that correlate with high Pbnf signal?

Thank you very much for your constructive suggestions. The statistical analysis in original Figure 1 was replaced by flow cytometry data that can accurately access the fluorescence emitted by each cell of double-labeled strains and determine their distribution patterns (revised Figure 1—figure supplement 1). Flow cytometry showed that in all double-labeled strains, mCherry fluorescence intensity generally correlated with GFP fluorescence intensity. For example, in strain *P_eps_*-*mCherry P_bnaF_*-*gfp*, the *P_eps_* signal was correlated with the *P_bnaF_* signal to a certain extent, highlighting the co-expression of EPS and BA synthesis genes in the same subpopulation (revised Figure 1—figure supplement 1). With regards to the experimental method of flow cytometry, (1) mild sonication of the biofilms was performed to segment cells for flow cytometry analysis; (2) the wild-type strain expressing no fluorescence protein was used as a negative control; (3) the negative control establishes the background fluorescence of the experimental samples and is used to set the threshold by adjusting baseline PMT (photomultiplier tube) voltages of the instrument.

For CLSM observation, the wild-type strain without fluorescent proteins was used as a negative control to estimate the fluorescence threshold. Besides, early-stage biofilms with monolayer cells were selected for observation.

2. For the data presented in Figure 2, how do the authors determine that the PI-stained cells were producers or cheaters before death? If a Peps-GFP cell dies, I presume it loses its GFP signal? Do the authors identify the cell types before death? Furthermore, in the figure legend, the authors should define what they mean by "near" an EPS- producing cell. Is this quantitatively defined?

Thank you for your comments. The SQR9-*P_eps_*-*gfp* cells emitting GFP signal are supposed to be producers because their EPS synthesis gene is activated; conversely, cells that don’t emit GFP signal are nonproducing cheaters. Based on this criterion, we determined the source of dead cells in the newly captured time-lapse images. The results showed that all dead cells were from non-*gfp* expressing cells (revised Figure 2—figure supplement 1), while the active *P_eps_*-*gfp* cells kept alive and didn’t lose their GFP signal during the cannibalism progress. Additionally, the distribution of dead cells in the newly captured time-lapse images were also analyzed. We define "near" an EPS- producing cell as cells that were in contact with an EPS-producing cell, also this information has been included in the figure legend. The number of dead cells adjacent to producers was significantly higher than that of non-adjacent cells (revised Figure 2—figure supplement 1).

3. The data in Figure 4E should be quantified, as similar data are in other figures.

Sorry for lack of quantification of data in Figure 4E. We have supplemented the statistical information (Duncan's multiple rang tests) in the revised Figure 4—figure supplement 3.